# Rac1 activates non-oxidative pentose phosphate pathway to induce chemoresistance of breast cancer

Qingjian Li[1,5], Tao Qin[1,5], Zhuofei Bi[1,5], Huangming Hong[1], Lin Ding[1], Jiewen Chen[1], Wei Wu[2,3], Xiaorong Lin[1], Wenkui Fu[1], Fang Zheng[3], Yandan Yao [2], Man-Li Luo[3], Phei Er Saw[3], Gerburg M. Wulf [4], Xiaoding Xu[3], Erwei Song [2,3], Herui Yao[1,3✉] & Hai Hu[1,3✉]

Resistance development to one chemotherapeutic reagent leads frequently to acquired tolerance to other compounds, limiting the therapeutic options for cancer treatment. Herein, we find that overexpression of Rac1 is associated with multi-drug resistance to the neoadjuvant chemotherapy (NAC). Mechanistically, Rac1 activates aldolase A and ERK signaling which up-regulates glycolysis and especially the non-oxidative pentose phosphate pathway (PPP). This leads to increased nucleotides metabolism which protects breast cancer cells from chemotherapeutic-induced DNA damage. To translate this finding, we develop endosomal pH-responsive nanoparticles (NPs) which deliver Rac1-targeting siRNA together with cisplatin and effectively reverses NAC-chemoresistance in PDXs from NAC-resistant breast cancer patients. Altogether, our findings demonstrate that targeting Rac1 is a potential strategy to overcome acquired chemoresistance in breast cancer.

[1] Department of Oncology, Sun Yat-Sen Memorial Hospital, Sun Yat-Sen University, 510120 Guangzhou, People's Republic of China. [2] Breast Tumor Center, Sun Yat-Sen Memorial Hospital, Sun Yat-Sen University, 510120 Guangzhou, People's Republic of China. [3] Guangdong Provincial Key Laboratory of Malignant Tumor Epigenetics and Gene Regulation, Sun Yat-Sen Memorial Hospital, Sun Yat-Sen University, 510120 Guangzhou, China. [4] Division of Hematology and Oncology, Beth Israel Deaconess Medical Center and Department of Medicine, Harvard Medical School, Boston, MA 02115, USA. [5] These authors contributed equally: Qingjian Li, Tao Qin, Zhuofei Bi. ✉email: yaoherui@163.com; huhai@mail.sysu.edu.cn

Chemotherapy is the standard cancer treatment which carries a pivotal role in cancer therapies. In particular, chemotherapy is required to treat patients of triple nega-tive, HER2-positive, or advanced luminal breast cancer when resistance occurs to endocrine therapy. Unfortunately, tumors will eventually develop chemoresistance despite the on-going treatment, rendering reduced effectiveness of chemotherapeutic agents[1]. Clinical observations have shown that once tumors become resistant to a certain drug in chemotherapy, they will rapidly acquire tolerance to other drugs, suggesting there are mechanisms involved in inducing multi-chemoresistance[2]. Although extensive work has been done to uncover the molecular mechanisms of chemoresistance, few has been successfully translated into clinical use. Therefore, there is a pressing need to identify key regulators and mechanisms critical to the develop-ment of chemoresistance, especially multidrug resistance, and to establish a reliable method for predicting and overcoming the chemoresistance.

Neoadjuvant chemotherapy (NAC) is widely used in cancer treatment. For breast cancer, NAC results in ~50% pathologic complete response (pCR) and partial response (PR), especially in triple-negative breast cancers (TNBC)[3]. As the NAC outcome can be clearly observed prior to surgery, the NAC setting presents an opportunity for studying chemoresistance in breast cancers. To explore the mechanisms of chemoresistance in breast cancer, we collect both NAC responsive and multi-drug resistant tumors. We also generate the breast cancer cell lines that are resistant to doxorubicin (DOX) and cisplatin (DDP). By analyzing mRNA expression profile, we identify that Rac1, a small GTP binding protein, is overexpressed in chemoresistant breast tumor tissues and cell lines. Inducing DNA damage among the major anti-cancer effects of chemotherapeutics[4,5]. Herein, we reveal that Rac1 acts as the key regulator to activate the glycolysis, especially the non-oxidative pentose phosphate pathway (PPP), to enhance the nucleotide metabolism and promote the repair of chemother-apeutics induced DNA damage. Knockdown of Rac1 expression decreases the resistance to chemotherapeutic drugs in breast can-cer cells and tumors. As there are no clinically available inhibitors for small GTP binding proteins, we develop an endosomal pH-responsive nanoparticle system to systematically deliver Rac1 siRNA together with cisplatin to breast tumors in vivo and indi-cating that Rac1 silencing effectively recovers the sensitivity to cisplatin in patient-derived xenografts (PDXs) derived from TNBC patients whose are resistant to platinum-based NAC. These studies suggest that targeting Rac1 represents a promising strategy to overcome multi-drug resistance to chemotherapy in breast cancers.

## Results

### Overexpression of Rac1 confers resistance to neoadjuvant chemotherapy.
To explore the key regulators of chemoresistance, we applied microarray to compare the expression profile of four NAC resistant and four NAC sensitive tumor tissues from triple negative breast cancer (TBNC) patients, who received DDP, docetaxel (DTX), and DOX-based treatment (Fig. 1a, Supple-mentary Table 1). Chemoresistance was defined as stable disease (SD) or progressive disease (PD) after NAC treatment, while chemosensitive patients had a completed response (CR) or partial response (PR) according to RECIST 1.1[6]. To explore the common mechanism that promote the chemoresistance in breast cancers, we generated doxorubicin-resistant MCF-7 cell line (MCF-7DR, Supplementary Fig. 1A, B) and compared the expression profile of MCF-7DR with its parental MCF-7 WT cells (Fig. 1b). A number of genes was commonly overexpressed or downregulated in both NAC chemoresistant tumors and MCF-7DR cells com-pared to the sensitive tumors and MCF-7 WT cells (Fig. 1c, d).

Among the 16 common overexpressed genes in both chemoresistant tumors and MCF-7DR cells, Rac1 overexpression was particularly noted, because Rac1 was not only expressed higher in breast tumors than in normal tissues (Supplementary Fig. 1C), but also its overexpression correlated with advanced tumor stage (Supplementary Fig. 1D) and poor survival in breast cancer patients (Supplementary Fig. 1E) in TCGA database. Furthermore, Rac1 was also upregulated in carboplatin resistant T47D cells (T47DCR) in comparison to T47D WT cells, besides in MCF-7DR cells in comparison to MCF-7 WT cells (Supple-mentary Fig. 1F–K), indicating the involvement of Rac1 in inducing resistance to multiple chemotherapiy drugs.

To further validate the expression of Rac1 in breast cancers, immunohistochemistry (IHC) was performed in 198 breast cancer tissues (Supplementary Table 2). We found that Rac1 was intensively expressed in breast cancer tissues compared to adjacent normal tissues (Fig. 1e, f and Supplementary Fig. 1L)[7]. In addition, high expression of Rac1 was associated with poor disease-free survival (DFS) (Fig. 1g) in the breast cancer patients. When analyzing the correlation of Rac1 expression with different subtypes of breast cancer, we discovered that the TNBC patients with high Rac1 level had the worst DFS and OS (Fig. 1h, i). In addition, multivariate Cox regression analyses demonstrated that Rac1 was an independent prognostic predictor for disease-free survival (DFS) ($p = 0.036$ for all breast cancer, and $p = 0.013$ for TNBC) (Supplementary Tables 4 and 5).

To further validate the correlation of Rac1 expression with breast cancer chemoresistance, we collected 133 core needle biopsies from breast cancer patients before neoadjuvant che-motherapy, which were then treated with platinum-based drugs, Taxanes, and Doxorubicin. Our clinical studies discovered that high Rac1 expression was correlated with poor outcome to neoadjuvant treatment in the breast cancer patients (Fig. 1j, k, Supplementary Table 3). Importantly, ROC curve showed that high Rac1 level significantly distinguished chemoresistant cases from sensitive cases of the breast cancers patients (Fig. 1l). These results suggested that high Rac1 level was associated with chemoresistance of breast cancers, and the expression level of Rac1 in tumor tissues can be used to predict the outcome of chemotherapy.

### Rac1 regulates chemosensitivity by influencing DNA damage repair.
We examined Rac1 expression in various breast cancer cell lines. Among these cells, MD-MBA-231 cells had relatively high Rac1 expression and showed less sensitivity to DDP, while MDA-MB-436 cells expressed relatively low Rac1 and were sensitive to DDP treatment (Supplementary Fig. 2A). Next, we manipulated Rac1 expression in MD-MBA-231 and MDA-MB-436 cells to explore the role of Rac1 in vitro. Rac1 is a key regulator of the actin cytoskeleton[8]. siRac1 (Fig. 2a) significantly suppressed migration and invasion of MD-MBA-231 cells (Supplementary Fig. 2B, C). However, our study showed that Rac1 expression did not correlate to lymph node or distant metastasis in breast cancer (Supplementary Table 2), implying the shorter DFS in Rac1 overexpressed breast cancers might be due to the resistance to chemotherapies (Fig. 1h and Supple-mentary Table 2).

Docetaxel, doxorubicin, and cisplatin are the first line agents of breast cancer chemotherapy in the neoadjuvant and adjuvant setting[9]. To test whether Rac1confered chemoresistance in breast cancer, we treated MD-MBA-231 cells with the above chemother-apeutic agents after Rac1 silencing. siRac1, which dramatically increased the sensitivity of MD-MBA-231 cells to these chemo drugs, as shown by the decreased cell proliferation (Fig. 2b), colony formation (Fig. 2c and Supplementary Fig. 2D) and

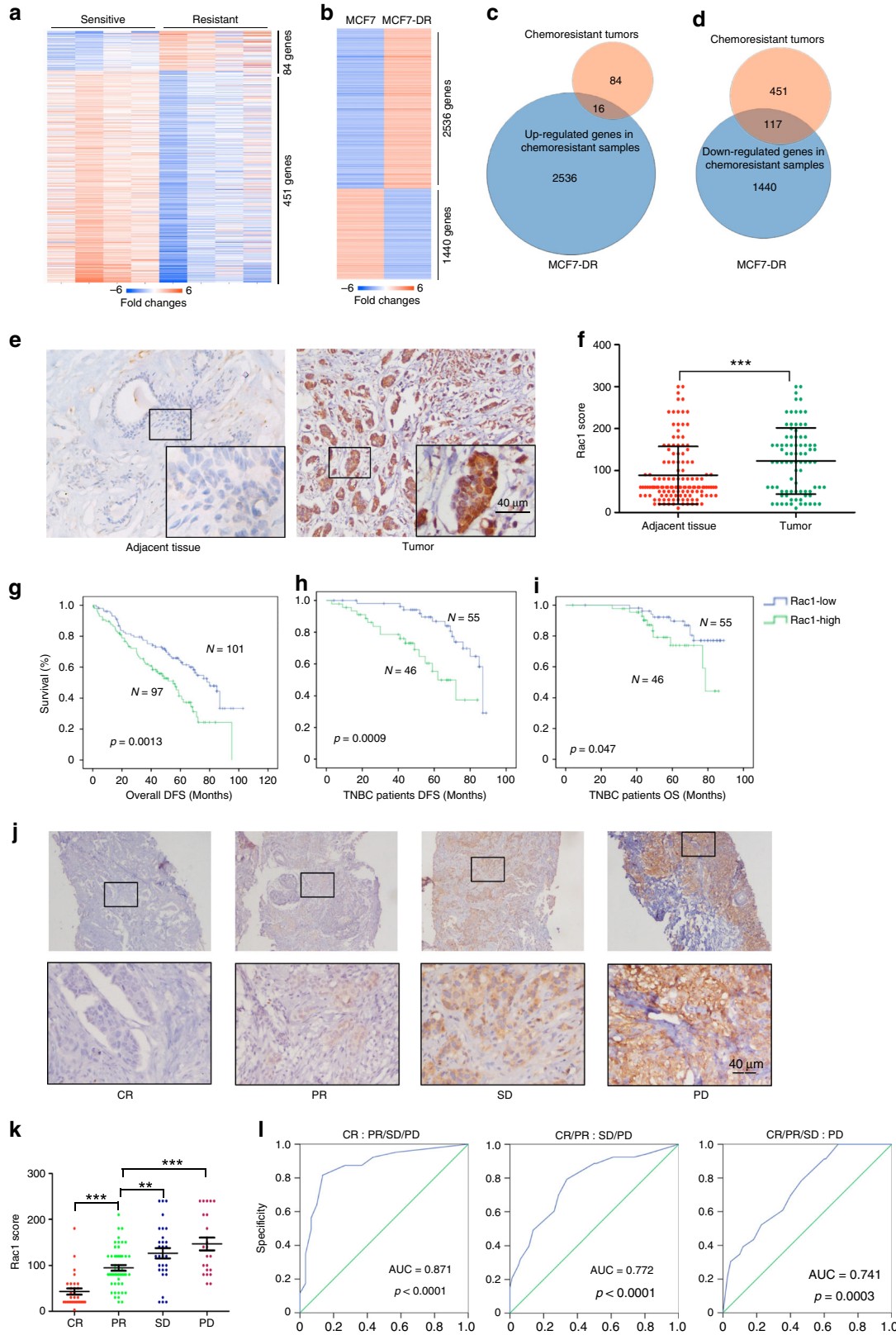

increased cell apoptosis upon drug treatment (Supplementary Fig. 2E, F). In addition, overexpression of Rac1 in MDA-MB-436 dramatically increased the resistance of MDA-MB-436 cells to these chemotherapeutic drugs, as shown by the increased cell proliferation, colony formation and reduced apoptosis (Fig. 2d–f, S2G–I).

Because we have shown that the doxorubicin-resistant MCF-7DR cells expressed high level of Rac1 (Supplementary Fig. 1F, G). We therefore silenced Rac1 expression in MCF-7DR cells and we found that the half maximal inhibitory concentration ($IC_{50}$) value of doxorubicin treatment significantly reduced almost 50% (Supplementary Fig. 3A). Silencing Rac1 also sensitized

**Fig. 1 Rac1 is upregulated in chemoresistant breast cancer and indicates worse prognosis and neoadjuvant chemotherapy outcome. a, b** The heatmaps of genes that were upregulated or downregulated over three folds in resistant tumor tissues (SD, PD, $n = 4$) comparing with sensitive tumor tissues (CR, PR, $n = 4$) of neoadjuvant chemotherapy (**a**) and genes that were upregulated or downregulated at least 6-fold in MCF-7DR comparing to that in parental MCF-7 (**b**). Expression levels were shown as log2 transformed intensity relative to the mean value of all samples. **c, d** The number of overlapping genes that were upregulated or downregulated among chemoresistant tumor samples and MCF-7DR in **a** and **b**. **e** Representative immunohistochemistry images of Rac1 expression in the breast cancer ($n = 198$) and adjacent normal tissues ($n = 86$). **f** Immunohistochemical staining scores of Rac1 in breast cancer (total $n = 198$, including 68 luminal, 29 HER2-positive and 101 TNBC patients) and adjacent tissues ($n = 86$). ($p < 0.0001$, two-sided unpaired $t$-test). **g–i** Higher levels of Rac1 in breast tumors were associated with poor disease-free survival (DFS) in all breast cancer patients ($n = 198$, $p = 0.0013$) (**g**), poor disease-free survival (DFS) (**h**) and overall survival (OS) (**i**) in TNBC patients. ($n = 101$, DFS $p = 0.0009$, OS $p = 0.047$. Kaplan–Meier, log-rank test). **j** Representative immunohistochemical images showed that Rac1 level was highest expressed in PD (progress disease) tumors ($n = 23$), following by that in SD (stable disease) ($n = 30$), PR (partial response) ($n = 50$) and CR (complete response) tumors ($n = 30$) of neoadjuvant chemotherapy treated breast cancer patients. **k** Immunohistochemical staining scores of Rac1 in CR ($n = 30$), PR ($n = 50$), SD ($n = 30$) and PD ($n = 23$) breast cancer patients treated with neoadjuvant chemotherapies ($n = 133$ in total, including 25 non-TNBC and 108 TNBC patients). (CR vs PR $p < 0.0001$, PR vs SD $p = 0.0088$, PR vs PD $p = 0.0002$, two-sided unpaired $t$-test). **l** ROC curve showed the diagnostic value of Rac1 in distinguishing neoadjuvant chemotherapy outcome of 133 breast cancer patients ($AUC_{CR} = 0.871$, $p < 0.0001$, $AUC_{CR/PR} = 0.772$, $p < 0.0001$, $AUC_{CR/PR/SD} = 0.741$, $p = 0.0003$, Mann–Whitney $U$ test). Bar graphs represent the mean ± SD of indicated samples. *$p < 0.05$, **$p < 0.01$, and ***$p < 0.001$. Source data are provided as a Source Data file.

MCF-7DR to doxorubicin treatment as shown by the reduction of colony formation (Supplementary Fig. 3B) and increased apoptosis (Supplementary Fig. 3C). Additionally, both Rac1-siRNA and Rac1 inactivation with NSC23766 inhibitor decreased the $IC_{50}$ of MD-MBA-231 cells upon cisplatin treatment (Supplementary Fig. 3D). On the other hand, exogenous expression of Rac1 increased the $IC_{50}$ doses of cisplatin in MDA-MBA-231 and MDA-MB-436 cells (Supplementary Fig. 3D, E). Together, these results suggested that Rac1 overexpression conferred resistance to chemotherapy in breast cancer cells.

The somatic mutation of *Rac1* has been discovered to function as a driving factor of malignancy in melanoma and other cancers[10–12]. Moreover, overexpression of Rac1 has been shown to be associated with poor outcome in several human cancers, such as breast cancer, colorectal cancer, and leukemia[13–16]. To test whether Rac1 affected the sensitivity of other cancer cells to chemotherapeutics, we manipulated Rac1 expression in the lung, ovary, and gastric cancer cell lines. The $IC_{50}$ of A549CR (carboplatin resistant lung cancer cell) decreased upon both Rac1 targeting siRNA and inhibitor treatment (Supplementary Fig. 3F). Rac1 overexpression in SKOV-3 (ovary cancer) and ASG (gastric cancer) increased its $IC_{50}$ dose to DDP treatment, whereas NSC23766 decreased the $IC_{50}$ doses in these cells (Supplementary Fig. 3G, H).

Since chemotherapeutic agents induce DNA damage directly or indirectly, DNA damage repairing ability profoundly affect the sensitivity of cancer cells to chemotherapies[4,5]. Thus, we examined whether Rac1 induced chemoresistance by enhancing DNA damage repair. Instead of chemotherapeutic agents, we used the sublethal ionizing radiation (IR) to induce DNA damage. Rac1 silencing suppressed the colony formation of MDA-MB-231, MCF-7DR, and T47DCR cells upon γ-irradiation (Fig. 2g–i), while overexpression of Rac1 in MDA-MB-436 cells increased the colony number upon γ-irradiation (Fig. 2j)[17]. Previous studies show that DNA damage is involved in cisplatin, doxorubicin and docetaxel induced tumor death[7,18,19]. We also found that γH2AX level was upregulated in siRac1 treated MDA-MB-231 cells and further increased by additional treatment of cisplatin, docetaxel or doxorubicin (Fig. 2k), while overexpression of Rac1 in MDA-MB-436 cells reduced the γH2AX level regardless with or without the treatment of these chemotherapeutic agents (Fig. 2l). Moreover, depletion of Rac1 by siRNA in MDA-MB-231 cells delayed DNA damage repair, while overexpression of Rac1 in MDA-MB-436 cells enhanced the DNA damage repair after IR treatment (Supplementary Fig. 3J–L). These results suggested that Rac1 promoted DNA damage repair to render cancer cells more resistance to chemotherapies.

**Rac1 activates non-oxidative pentose phosphate pathway**. Cell metabolism plays a fundamental role in regulating cancer progression as well as their resistance to chemotherapies[20,21]. Gene Set Enrichment Analysis[22] of the mRNA expression profiles of the NAC treated TNBCs revealed that dysregulation of metabolic pathway was one of the major changes between chemosensitive and chemoresistant breast cancers (Fig. 1a and Supplementary Table 6). Therefore, we screened the metabolites by mass spectrometry in Rac1 knockdown cells (Supplementary Fig. 4A) and found in doxycycline (doxy)-inducible Rac1 knockdown (Plko-tet-on) MDA-MB-231 cells, the metabolites of upper glycolysis and non-oxidative pentose phosphate pathway were decreased upon shRac1 (Fig. 3a, b). Consistently, we found that the knockdown of Rac1 resulted in the decreased glucose uptake in MDA-MB-231 cells, while Rac1 overexpression increased the glucose uptake in MDA-MB-436 cells (Fig. 3c).

Aldolase has been reported as a cytoskeleton binding enzyme[23–25]. We and others have shown that the cytoskeleton binding of aldolase A restricts its enzymatic activity[23,26–28]. PI3K accelerates the cytoskeleton turnover through activating Rac1, thus releases aldolase A from cytoskeleton to become the cytosolic soluble and active fraction[28,29]. Furthermore, inhibition of PI3K-Rac1-cytoskeleton axis in breast tumors results in limited supply of ribose, leading to DNA damage and enhanced tumor sensitivity towards PARP inhibitors[29]. To test whether Rac1 overexpression influences aldolase activity, we permeabilized MCF-7 WT and MCF-7DR cells with digitonin to allow the efflux of cytosolic soluble aldolase A diffusing into supernatant[28], and then separately collected the supernatant and the cell lysate for immunoblotting. We found that the level of supernatant aldolase A was higher in MCF-7DR than that in MCF-7 WT cells (Fig. 3d). The transient transfection of GFP-Rac1 resulted in an increased level of aldolase A in the supernatant of MDA-MB-436 cells (Fig. 3e). In line with these observations, knockdown of Rac1 in MDA-MB-231 and MCF-7DR cells decreased aldolase A level in the supernatant (Fig. 3f and Supplementary Fig. 4B), while cisplatin or doxorubicin treatment did not significantly affect the level of cytoskeleton-free aldolase A (Fig. 3f and Supplementary Fig. 4B). In addition, Rac1 inhibition with NSC23766 resulted in the decreased level of cytoskeleton-free aldolase A in MDA-MB-231 cells (Supplementary Fig. 4C). Consistently, overexpressing Rac1 in MDA-MB-436 cells or knockdown of Rac1 in MDA-MB-231 cells resulted in the increased or decreased aldolase activity, respectively (Fig. 3g).

Moreover, we found that phosphorylated-ERK (p-ERK) level was increased along with the overexpression of Rac1 in MCF-7DR cells as compared to parental MCF-7 WT cells (Fig. 3d).

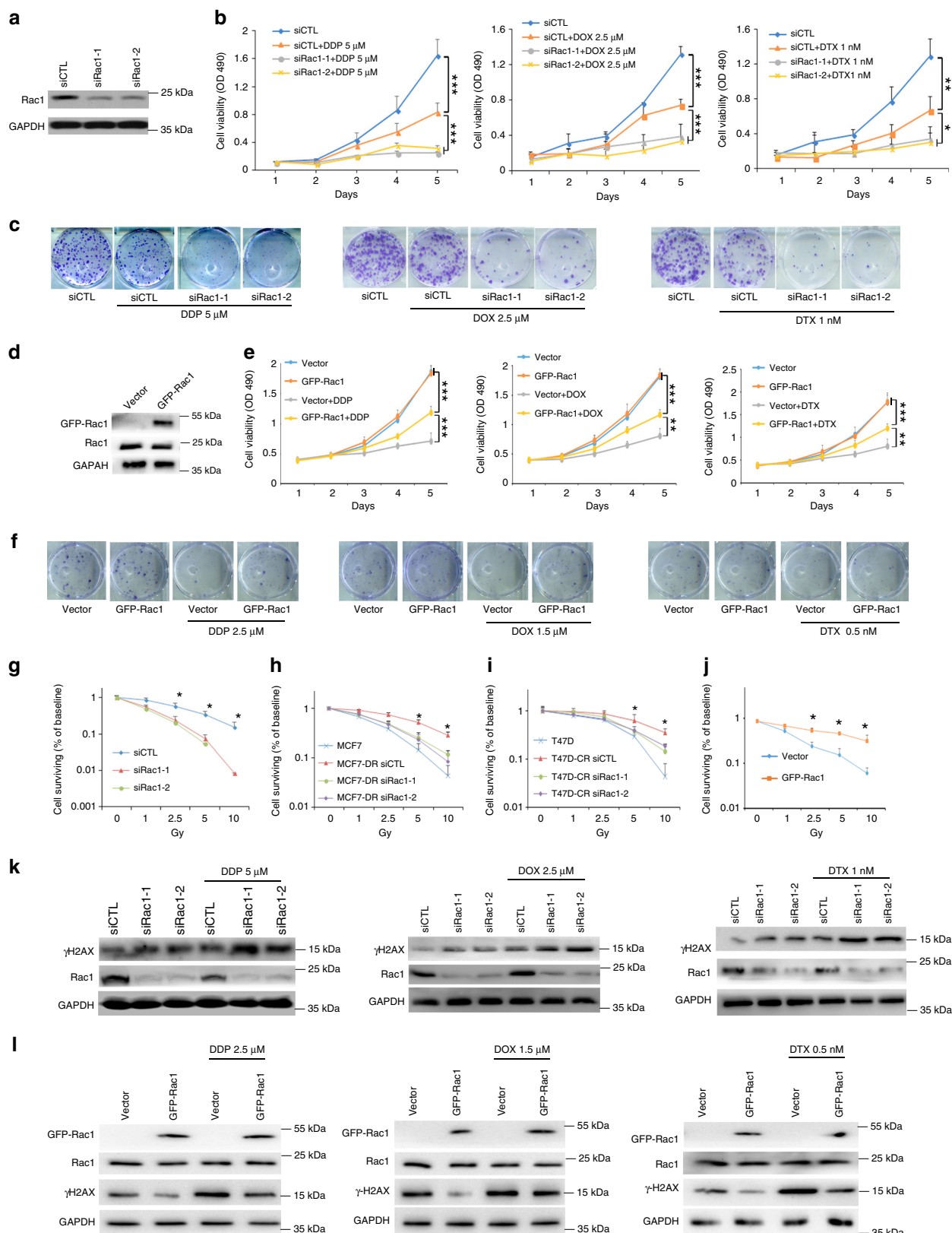

Manipulation of Rac1 level in MDA-MB-436, MDA-MB-231, and MCF-7DR cells revealed changes of p-ERK level along with Rac1 level (Fig. 3e, f and Supplementary Fig. 4B). Consistently, Rac1 inhibition with NSC23766 decreased p-ERK level in MDA-MB-231 cells (Supplementary Fig. 4C). To explore whether Rac1 regulated aldolase activity via ERK signaling, we treated MDA-

MB-231 cells with an ERK inhibitor (SCH772984) and found that ERK inhibition did not affect the level of cytoskeleton-free aldolase in MDA-MB-231 cells, suggesting that Rac1 regulated aldolase activity in an ERK-independent manner (Fig. 3f). In addition, we found that the MAPK pathway was activated and the non-oxidative PPP pathway enzymes were overexpressed in the

**Fig. 2 Rac1 knockdown increases the chemosensitivity of breast cancer cells by inducing DNA damage. a** Rac1 protein level of MDA-MB-231 decreased after transient transfection of siRac1, as detected by western blots. **b** Rac1 knockdown increased the chemosensitivity of MDA-MB-231, as detected by MTS assay. Cells were treated with 5 µM DDP, 2.5 µM doxorubicin, or 1 nM docetaxel. **c** Rac1 knockdown increased the chemosensitivity of MDA-MB-231, as detected by colony formation assay. The result was obtained over three independent experiments. Also see Supplementary Fig. 2D. **d** GFP-Rac1 protein level in MDA-MB-436 after transient transfection of exogenous GFP-Rac1, as detected by western blots. **e** Rac1 overexpression increased the chemoresitance of MDA-MB-436, as detected by MTS assay. Cells were treated with 2.5 µM DDP, 1.5 µM doxorubicin, or 0.5 nM docetaxel. **f** Rac1 overexpression increased the chemoresitance of MDA-MB-436, as detected by colony formation assay. The result was obtained over three independent experiments. Also see Supplementary Fig. 2G. **g–i** Silencing Rac1 increased the sensitivity of breast cancer cells to radiation. MDA-MB-231 (**g**), MCF-7DR (**h**) and T47DCR (**i**) were treated with different dose of ionizing radiation (0–10 Gy) after transient transfection with siRNAs for 48 h. **j** Overexpressing Rac1 decreased the sensitivity of breast cancer cells to radiation. MDA-MB-436 were treated with different dose of ionizing radiation (0–10 Gy) after transient transfection with GFP-Rac1. (2.5 Gy $p = 0.0065$, 5 Gy $p = 0.0025$, 10 Gy $p = 0.0154$, two-sided unpaired $t$-test). **k** Silencing Rac1 further increased the level of γH2ax of MDA-MB-231 cells treated with chemotherapeutic drugs. Cells was treated with 5 µM DDP, 2.5 µM doxorubicin, or 1 nM docetaxel for 24 h after transiently transfected with siRNAs. **l** Overexpressing Rac1 decreased the level of γH2ax of MDA-MB-436 cells treated with chemotherapeutic drugs. Cells was treated with 2.5 µM DDP, 1.5 µM doxorubicin, or 0.5 nM docetaxel for 24 h after transiently transfected with GFP-Rac1. The result of immunoblotting was obtained over three independent experiments. Bar graphs represent the mean ± SD of experimental triplicates. *$p < 0.05$, **$p < 0.01$ and ***$p < 0.001$. Source data are provided as a Source Data file.

chemoresistant TNBCs and MCF-7DR cells (Supplementary Table 6 and Supplementary Fig. 4D, E) as compared with chemosentivtive TNBCs and MCF-7 WT respectively. Therefore, we further examined whether the levels of non-oxidative PPP pathway enzymes were regulated by Rac1-MAPK axis. Depletion of Rac1 in MDA-MB-231 decreased the expression of non-oxidative PPP pathway enzymes (Fig. 4a), whereas overexpressing Rac1 in MCF-7 cells resulted in the upregulation of non-oxidative PPP pathway enzymes, which could be reversed by ERK inhibition (Fig. 4b).

P21-activated protein kinase (PAK) is one of the major downstreasm targets of Rac1[30], which phosphorylates RAF and subsequently activates MEK/ERK pathway[12,31,32]. We found that siRac1 reduced the p-PAK1/2 level in MDA-MB-231 cells and overexpression of Rac1 augmented PAK1/2 phosphorylation in MDA-MB-436 cells (Fig. 4c). In addition, silencing PAK1 could abolish Rac1 mediated activation of the RAF/MERK/ERK signaling cascade (Fig. 4d), supporting that Rac1 activated the RAF/MEK/ERK pathway through its downstream effector PAK. Furthermore, luciferase assay revealed that the transcriptional activity of non-oxidative PPP pathway enzymes, Rpia and Tkt, was elevated upon Rac1 overexpression and could be suppressed by the additional application of ERK inhibitor (Fig. 4e). These data suggested that Rac1 enhanced the transcription of the non-oxidative PPP pathway enzymes via activation of its downstream PAK/RAF/MERK/ERK signaling cascade.

To confirm whether non-oxidative PPP flux was regulated by Rac1, we applied carbon tracing to follow the fate of radioactive-labeled glucose which would be incorporated into the deoxyribose–containing backbone of newly synthesized DNA. If 6-$^{14}$C glucose is used for ribose synthesis, the $^{14}$C ribose can be metabolized either through the oxidative or the non-oxidative arm of the pentose phosphate pathway (PPP). On the other hand, in oxidative PPP pathway, the radioactive label of the 1-$^{14}$C glucose is lost as $CO_2$ and will only preserve if ribose is produced via the non-oxidative PPP[33]. Therefore, we first treated MDA-MB-231 cells with 6-$^{14}$C glucose or 1-$^{14}$C glucose and revealed that both 1-$^{14}$C and 6-$^{14}$C radiation was reduced after Rac1 silencing, while the 1-$^{14}$C labeling was more significantly reduced as compared to the 6-$^{14}$C labeling in MDA-MB-231 cells (Fig. 4f). These observations supported that Rac1 regulated the ribose synthesis mainly through non-oxidative PPP. Consistently, siRac1 resulted in the reduction of the nucleosides pool in MDA-MB-231 cells (Fig. 4g). Together, these results supported that Rac1 enhanced the glycolysis and non-oxidative PPP to promote ribose synthesis, and knocking down Rac1 would restrict the ribose supply and compromise the DNA damage repair.

**Repletion of nucleosides rescues DNA damage caused by Rac1 knockdown.** We have shown that Rac1 enhanced glycolysis and non-oxidative PPP to increase ribose synthesis through activation of the aldolase and ERK signaling. To confirm the hypothesis that Rac1 induced chemoresistance by increasing ribose synthesis to enhance DNA damage repair upon chemotherapeutic treatment, we tested whether the siRac1-induced DNA damage could be rescued by exogenous aldolase or ERK. As binding with cytoskeleton would restrict the enzymatic activity of aldolase[23,26–28], we transfected MDA-MB-231 ells with aldolase A R42A mutant, which was shown to be unable to bind cytoskeleton yet maintaining catalytic activity[27]. Because the WT aldolase A would spontaneously bind to cytoskeleton as we had desribed before[28], the aberrant expression of aldolase A R42A, but not WT aldolase A, increased the whole enzymatic activity of aldolase in the MDA-MB-231 cells (Fig. 5a). We found that the siRac1-induced DNA damage, as detected by elevated γH2AX level, could be partially rescued by exogenously expressing aldolase A R42A or ERK1, and almost completely reduced by the combination of R42A aldolase A and ERK1 overexpression (Fig. 5b).

As siRac1 treatment resulted in reduced nucleosides pool (Fig. 4c), we hypothesized that direct repletion of nucleosides (Adenosine, Guanosine, Cytidine, Uridine, A, G, C, U) should rescue the DNA damage induced by siRac1. Indeed, siRac1-induced DNA damage, as shown by γH2AX levels, was completely rescued by adding nucleosides (A, G, C, U; 100 µM respectively) to the culture medium of MDA-MB-231 cells (Fig. 5c). Moreover, cisplatin-induced aggravated DNA damage could also be rescued partially by nucleosides repletion in siRac1 MDA-MB-231 cells, as measured by the γH2AX level (Fig. 5d), cell viability (Fig. 4e), colony formation (Fig. 5f and Supplementary Fig. 4F) and apoptosis (Fig. 5g and Supplementary Fig. 4G). These data confirmed that Rac1 enhanced the glycolysis and non-oxidative PPP to promote the ribose synthesis. Knockdown of Rac1 would restrict the ribose supply and thus compromised the DNA damage repair.

**Rac1 knockdown increases the sensitivity of breast tumors to chemotherapies.** To examine the role of Rac1 in chemoresistance in vivo, we injected inducible control shRNA or shRac1 (Plko-tet-on) expressing MDA-MB-231 cells into the mammary pad of nude mice. Once the average volumes of xenografts in each group reached ~150 mm³, the mice were fed with doxycycline to induce shRNA expression and treated with cisplatin (4 mg kg$^{-1}$ weekly). Rac1 knockdown slightly deceased the tumor growth, while cisplatin treatment moderately decreased the tumor growth in vivo

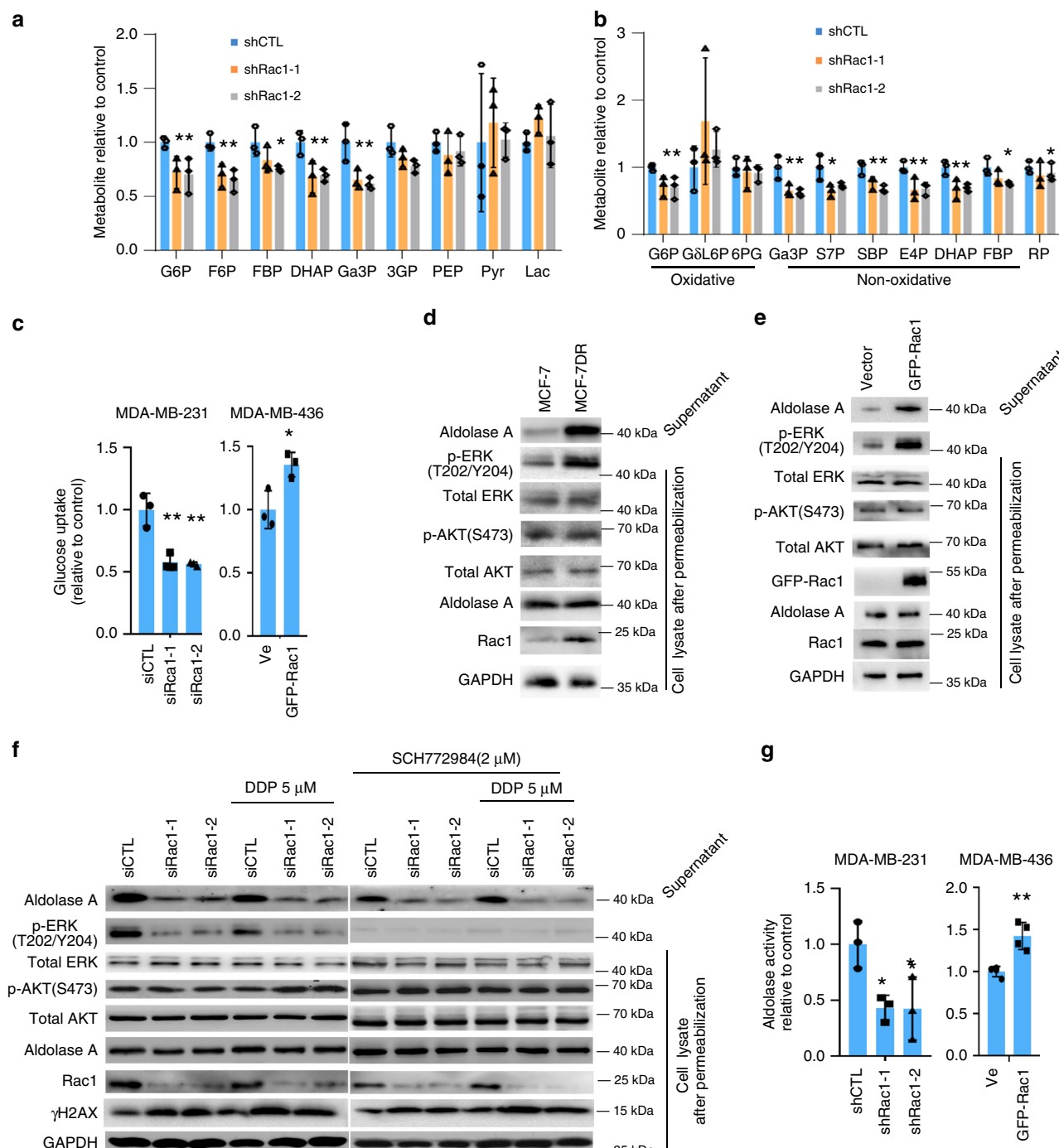

**Fig. 3 Rac1 regulates glycometabolism and non-oxidative pentose phosphate pathway (PPP) via affecting aldolase activity. a**, **b** The levels of upper glycolysis metabolites (**a**) and the glycolytic intermediates of non-oxidative PPP (**b**) decreased upon Rac1 knockdown in MDA-MB-231 cells. All metabolite levels were normalized to the vehicle control. Bar graphs represent the mean ± SD of experimental triplicates. **c** Glucose uptake decreased upon Rac1 knockdown and increased following Rac1 overexpression. (MDA-MB-231 siCTL vs si-1 $p = 0.0079$, siCTL vs si-2 $p = 0.00475$, MDA-MB-468 Vector vs GFP-Rac1 $p = 0.0258$, two-sided unpaired $t$-test) Bar graphs represent the mean ± SD of experimental triplicates. **d** Rac1 level associated with aldolase A level in supernatant and p-ERK level in cell lysate of MCF7 and MCF7R cells. The result was obtained over three independent experiments. **e** Rac1 overexpression increased the level of aldolase in supernatant and p-ERK in cell lysate of MDA-MB-436 cells. The result was obtained over three independent experiments. **f** Rac1 knockdown but not Erk inhibition reduced the level of aldolase in supernatant of MDA-MB-231 cells. ERK inhibitor SCH772984 was used. The result was obtained over three independent experiments. **g** Rac1 level associated with aldolase activity in breast cancer cells. (MDA-MB-231 siCTL vs si-1 $p = 0.0145$, siCTL vs si-2 $p = 0.0469$, MDA-MB-436 Vector vs GFP-Rac1 $p = 0.0026$, two-sided unpaired $t$-test). Bar graphs represent the mean ± SD of experimental triplicates. $*p < 0.05$, $**p < 0.01$, and $***p < 0.001$. Source data are provided as a Source Data file.

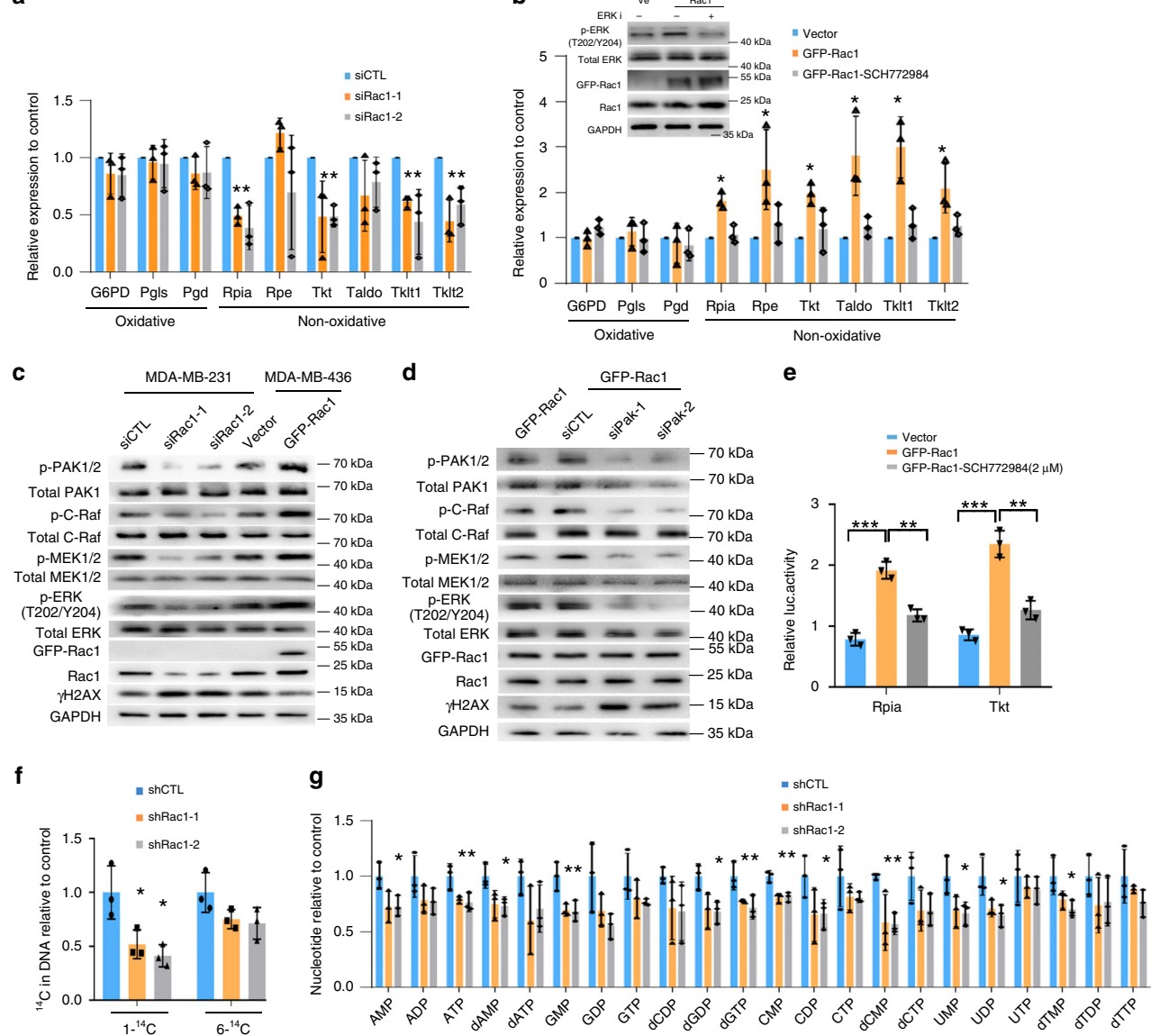

**Fig. 4 Rac1 regulates non-oxidative pentose phosphate pathway via ERK signaling. a** Rac1 knockdown decreased mRNA levels of PPP enzymes in MDA-MB-231 cells. Expression of PPP enzymes were quantified by qRT-PCR, and normalized to GAPDH expression in siCTL cells. Bar graphs represent the mean ± SD of three independent experiments. **b** Exogenous expression of Rac1 increased the level of PPP enzymes, which could be reversed by ERK inhibition in MCF-7 cells. The mRNA levels of PPP enzymes were quantified by qRT-PCR. Bar graphs represent the mean ± SD of three independent experiments. *$p < 0.05$. **c** Manipulating Rac1 expression affected PAK1/2 phosphorylation and the c-Raf/MEK/ERK signaling in breast cancer cells. The result was obtained over three independent experiments. **d** Exogenous expression of GFP-Rac1 activated c-Raf/MEK/ERK cascade, which could be reversed by PAK1 knockdown. MDA-MB-436 cells were transiently transfected with GFP-Rac1 or siPAK1. The result was obtained over three independent experiments. **e** Luciferase assay revealed that the transcription of PPP enzymes Rpia and Tkt was activated by Rac1 overexpression, which could be reversed by ERK inhibition. MDA-MB-436 cells with exogenous GFP-Rac1 were used. (Rpia Vector vs GFP-Rac1 $p = 0.0003$, GFP-Rac1 vs SCH772984 $p = 0.0016$, Tkt Vector vs GFP-Rac1 $p = 0.0004$, GFP-Rac1 vs SCH772984 $p = 0.0022$, two-sided unpaired $t$-test). Bar graphs represent the mean ± SD of three independent experiments. **f** Rac1 knockdown reduced the carbon flux of non-oxidative PPP. Carbon flux from glucose to ribose was determined by 14C-glucose derived carbon incorporated DNA in MDA-MB-231 cells. 14C-uptake in shRac1 cells was displayed, with experimental triplicates ± SD, normalized to 14C-uptake in control cells. Bar graphs represent the mean ± SD of experimental triplicates. **g** Silencing Rac1 caused a reduction of nucleotides pool. The MDA-MB-231 cells were treated as in **a**, and the lysis was prepared for mass spectrometry analysis. Bar graphs represent the mean ± SD of experimental triplicates. *$p < 0.05$, **$p < 0.01$, and ***$p < 0.001$. Source data are provided as a Source Data file.

(Fig. 6a–c). However, when shRac1 and cisplatin were treated in combination, there was a dramatic synergistic effect that led to almost total regression of tumors in some of the mice (Fig. 6a–c). IHC staining of the tumor sections confirmed the decreased level of Rac1 and p-ERK in shRac1 tumors (Fig. 6d, e and Supplementary Fig. 5A). We also found that the level of γH2AX was increased in shRac1 or cisplatin treated tumors, and was further elevated by the

combination of shRac1 and cisplatin (Fig. 6f and Supplementary Fig. 5B). Ki67 and cleaved Caspase-3 (CC3) staining also confirmed the decreased proliferation and increased apoptosis in the xenografts (Fig. 6g, h and Supplementary Fig. 5C, D).

The MCF-7 WT and MCF-7DR cells with or without inducible Rac1 knockdown (Plko-tet-on) were also used for xenograft experiments. Nude mice were implanted with 1.7 mg 17β-estradiol

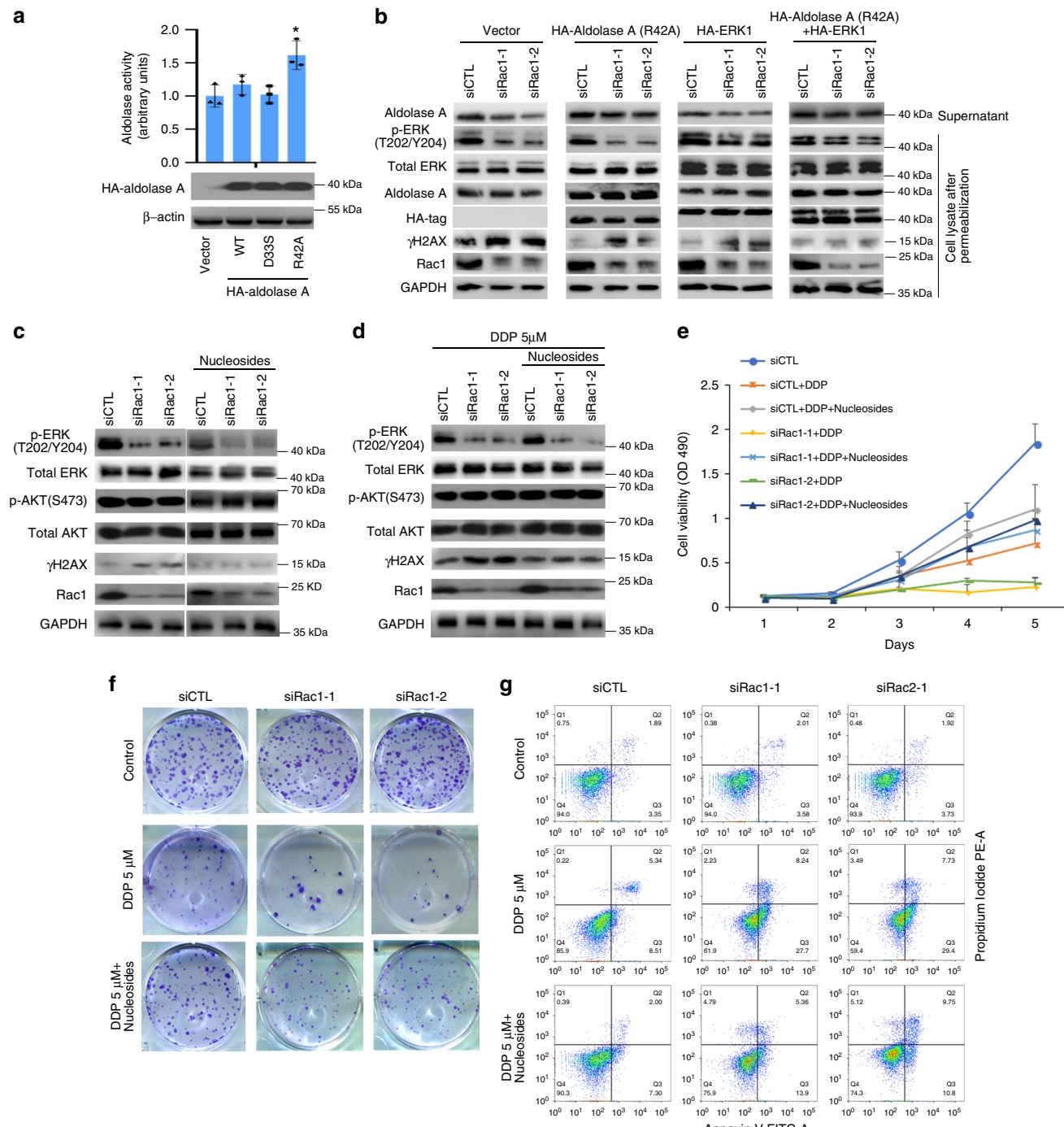

**Fig. 5 Rac1 knockdown induced DNA damage can be rescued by exogenous expression of aldolase, ERK1 and nucleosides repletion. a** Exogenous expression of aldolase A mutant (R42A) significantly increased aldolase enzyme activity. Exogenous expression of wild-type and mutant aldolase A in MDA-MB-231 cells were detected by western blots (lower panel). The aldolase activity was determined by the enzyme assay (upper panel). The result was obtained over three independent experiments. Bar graphs represent the mean ± SD of experimental triplicates. *$p < 0.05$. **b** Effects of exogenous expressed HA-aldolase A (R42A) or HA-ERK1 on abrogating γH2ax upregulation induced by Rac1 knockdown. MDA-MB-231 cells were transfected with empty vector, HA-aldolase A (R42A) or HA-ERK1 after transiently transfected with siCTL, siRac1-1, and siRac1-2 for 48 h, washed with cold PBS and then permeabilized with 30 μg/ml digitonin/PBS for 5 min at 4 °C . Then the supernatant was collected and the cells were lysed for immunoblotting. The result was obtained over three independent experiments. **c, d** Effects of nucleosides repletion on abrogating γH2ax upregulation induced by Rac1 knockdown with (**c**) or without DDP treatment (**d**). MDA-MB-231 cells were transiently transfected with siRNAs for 48 h and then treated with or without 5 μM DDP for 24 h in the cultured medium with additional indicated nucleosides (**a**, **g**, **c**, **u**; 100 μM). The result was obtained over three independent experiments. **e–g** Nucleosides rescued siRac1-induced phenotypes upon DDP treatment. MTS assay (**e**), colony formation assay (**f**), and apoptosis assay (**g**) were performed to evaluate the effects of nucleosides repletion (**a**, **g**, **c**, **u**; 100 μM) on rescuing Rac1 silencing phenotypes upon DDP treatment. Bar graphs represent the mean ± SD of experimental triplicates (**e**, **f**). Bar graphs represent the mean ± SD of three independent experiments (**g**). Also see Supplementary Fig. 4E, F. Source data are provided as a Source Data file.

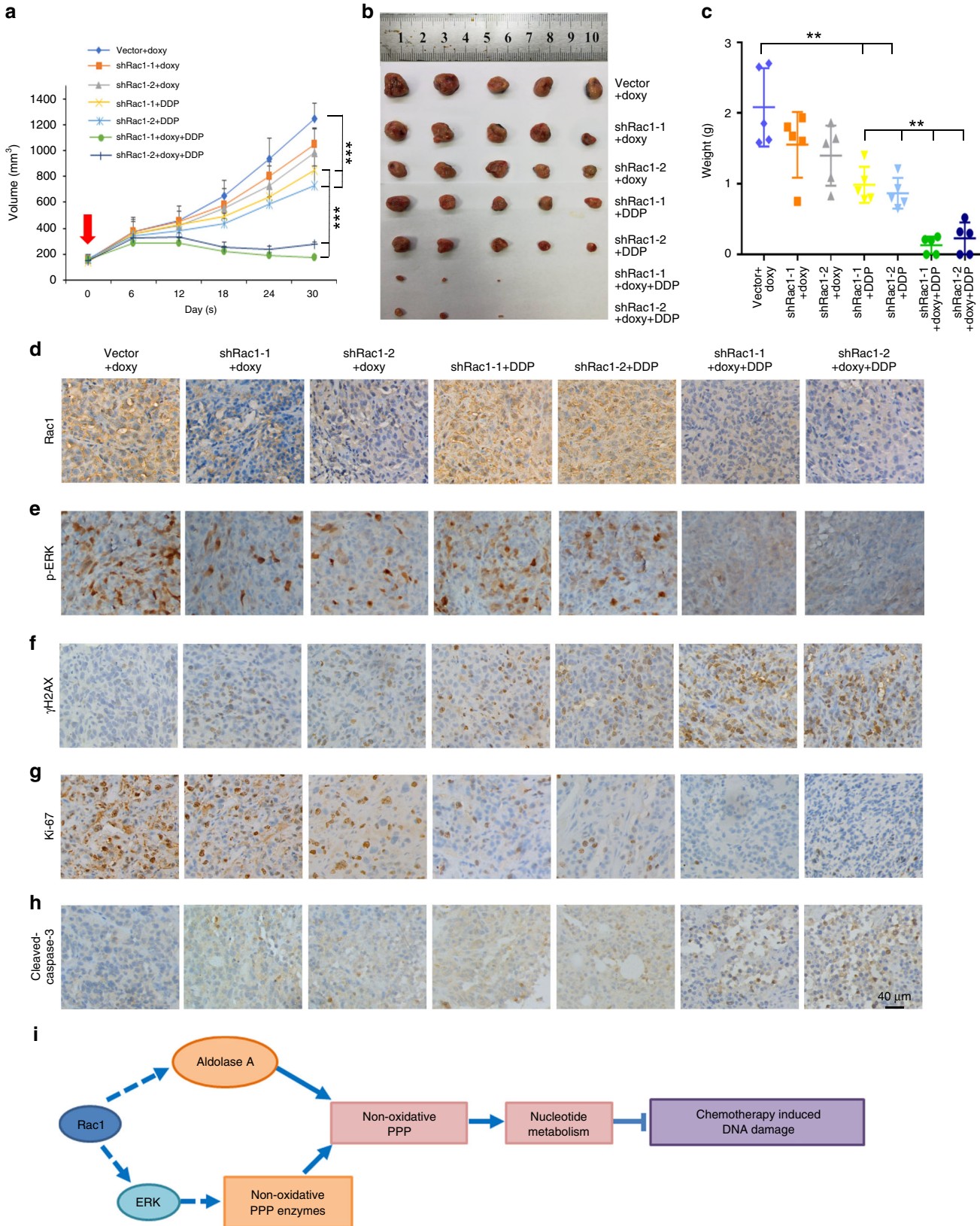

pellets (60-day release) 3 days before the mammary fat pad inoculation of MCF-7 WT or MCF-7DR cells as described before[34]. The mice were then fed with doxycycline and treated with doxorubicin (2 mg kg$^{-1}$ weekly) intraperitoneally once the average tumor volume reached ~150mm$^3$. Our result showed that the knockdown Rac1 moderately decreased the growth of MCF-7DR tumors, while the doxorubicin treatment only slightly reduced the tumor growth of MCFDR (Supplementary Fig. 5E±G).

Strikingly, combination treatment of doxorubicin and Rac1 depletion, most significantly decreased the MCF7DR tumor growth (Supplementary Fig. 6E–G). The down-regulation of Rac1 and p-ERK levels in the shRac1 tumors was confirmed by

**Fig. 6 Rac1 knockdown increases chemosensitivity and reduced chemotherapy resistance of breast tumors. a–c** Growth curve (**a**), tumor image (**b**), and tumor weights (**c**) of MDA-MB-231 xenografts treated with DDP. MDA-MB-231 transfected with control shRNA (as Vector) or Plko-tet-on-shRac1 (as shRac1) were injected into mammary fat pad of nude mice. When the tumor size reached ~150 mm$^3$, mice were injected with DDP intraperitoneally (4 mg/kg weekly), and fed with doxycycline (doxy) (2 mg/ml) to induce Rac1 knockdown (indicated by the arrow). Xenografts ($n = 5$ per group) were harvested 30 days post injection. (Growth curve Vector vs sh-1+DDP $p < 0.0001$, Vector vs sh-2+DDP $p < 0.0001$, sh-1+DDP vs sh-1+DDP + doxy $p < 0.0001$, sh-2 +DDP vs sh-2+DDP + doxy $p < 0.0001$, two-way ANOVA + Dunnett's post hoc tests, Tumor weight Vector vs sh-1+DDP $p = 0.0039$, Vector vs sh-2 +DDP $p = 0.0018$, sh-1+DDP vs sh-1+DDP + doxy $p = 0.0001$, sh-2+DDP vs sh-2+DDP + doxy $p = 0.0019$, two-sided unpaired $t$-test). Also see Supplementary Fig. 5A. **d–h** Representative immunohistochemical images of paraffin-embedded xenograft sections. Rac1 (**d**), p-ERK (**e**), γH2ax (**f**), Ki67 (**g**), and cleaved caspase-3 staining. Each group $n = 5$. Scale bar, 40 μM. **h** were scored from five randomly chosen fields from different tumors or each group. Scale bar, 20 μm. Also see Supplementary Fig. 5B–D. **i** A schematic model showed that Rac1 enhanced the activity of non-oxidative pentose phosphate pathway to inhibit the DNA damage caused by chemotherapeutics in breast cancer. Data are presented as mean ± SD of indicated samples. *$p <$ 0.05, **$p < 0.01$, and ***$p < 0.001$. Source data are provided as a Source Data file.

IHC staining (Supplementary Fig. 6A, B). Similarly, the levels of γH2AX, cleaved Caspase-3 and Ki67 staining indicated an increased in DNA damage and cell apoptosis as well as reduced cell proliferation in tumors treated with the combination of shRac1 and doxorubicin (Supplementary Fig. 6C–E). Together, these results suggested that silencing Rac1 enhanced the chemosensitivity of breast tumors.

**Systemic delivery of Rac1 siRNA by nanoparticles recovers the chemosensitivity of breast tumors.** Rac1 is a member of the small guanosine triphosphatases (GTPases), which was considered "hard-to-target" despite numerous efforts to develop GTPase inhibitors[35]. Recently, targeting mRNA instead of protein by antisense or siRNA oligonucleotides has become an alternative strategy to target GTPase for cancer treatment[36,37]. However, siRNAs, the polyanionic biomacromolecules, are easily attacked by serum nucleases and cannot readily cross the cell membrane. Therefore, specific delivery vehicles are required to facilitate the cytosolic siRNA delivery[38–40]. In recent years, nanoparticles (NPs) have been demonstrated as powerful tools for systemic siRNA delivery and several RNAi NP platforms have entered into early phase clinical trials for the treatment of various diseases including cancer[41–43]. This NP is made with an endosomal pH-responsive methoxyl-poly (ethylene glycol)-*b*-poly(2-(diisopropylamino) ethyl methacrylate) (Meo-PEG-b-PDPA) polymer with a p$K_a$ (~6.24) close to the endosomal pH (6.0–6.5) When using this NP platform for siRNA delivery, it can rapidly respond to endosomal pH and efficiently escape from endosomes via "proton sponge" effect to improve gene silencing efficacy (Fig. 7a)[44–46]. We have developed a RNAi NP platform for an efficient siRNA delivery and promising anticancer effect in vivo[47,48]. To evaluate whether the systemic delivery of siRac1 could silence Rac1 expression in vivo and thereby improve the efficacy of chemotherapeutics (e.g., DDP), we employed this NP platform to concurrently deliver siRac1 and DDP. Given by the fact that DDP was a hydrophilic drug which could not be efficiently encapsulated into NPs, we therefore incorporated two hydrophobic tails to its structure (denoted DDP prodrug) to enhance its encapsulation efficiency (Fig. 7a, Supplementary Fig. 8). For this DDP prodrug, the two hydrophobic tails could be cleaved by reductive agents such as glutathione (GSH) in the cytoplasm[49–51], and thus intact DDP would be produced.

First, we tested the physiochemical properties of the NPs loading with siRac1 and DDP prodrug (denoted siRac1/DDP NPs, siRNA NPs and DDP NPs) (Fig. 7). The well-defined spherical siRac1/DDP NPs could be formed with an average size of around 30 nm (Fig. 7b, c). The average size of siRNA NPs and DDP NPs were ~20 nm (Supplementary Fig. 6F–I). In this self-assembly system, the siRNA and DDP prodrug could be concurrently encapsulated into the NPs made with the Meo-PEG-*b*-PDPA polymer[52,53]. By using the fluorescent dye Cy5 to label the siRac1 (denoted Cy5-siRac1), the encapsulation efficiency of siRNA was determined as

~80%. Similar to our previous studies, due to the pH-responsive characteristic of Meo-PEG-*b*-PDPA polymer, the resulting Cy5-siRac1/DDP NPs showed pH-dependent cargo release behavior. With the protonation of the Meo-PEG-*b*-PDPA polymer to induce the disassembly of the NPs, more than 80% of the loaded siRNA or DDP prodrug was released within 12 h at a pH of 6.0 (Fig. 7d). Within the same time frame, less than 40% of the loaded cargos was released at a pH of 7.4 (Fig. 7d). More importantly, the protonation of the Meo-PEG-*b*-PDPA polymer could improve the endosomal escape ability of the NPs via the so-called "sponge" effect[44,45]. After incubating the Cy5-siRac1/DDP NPs with MDA-MB-231 cells for 4 h, a majority of the internalized siRNAs (red fluorescence) escaped from the endosomes (green fluorescence) and entered the cytoplasm where siRNA functioned (Fig. 7e). On the other hand, the non-pH-responsive NPs which was prepared by the commercially available polymer, Meo-PEG-b-PLGA, was used to load the Cy5-siRac1/DDP as negative control. After incubating the Cy5-siRac1/DDP NPs (non-pH-responsive) with MDA-MB-231 cells for 4 h, the majority of the internalized siRNAs (red fluorescence) could not escaped from the endosomes (green fluorescence) (Fig. 7e). With this promising endosomal escape function, the siRac1/DDP NPs efficiently suppressed Rac1 expression in the MDA-MB-231 cells, and there was nearly no Rac1 expression at a 40-nM siRNA dose (Fig. 7f, g). As a consequence, the loaded DDP with increased dose of siRac1 prodrug induced severe DNA damage as demonstrated by the increased level of γH2AX expression (Fig. 7f). Coincidently, the siRNA we used could also target the murine Rac1. Thus, we applied the siRac1/DDP NPs to mouse breast cancer cell 4T1, and found the similar effect of suppressing mouse Rac1 expression, as well as inducing DNA damage (Supplementary Fig. 6J, K). Therefore, this siRNA was ideally to assess the therapeutic efficacy and safety of siRac1/DDP NPs in the mouse model bearing human tumors. To evaluate the synergistic effect of siRac1 with DDP, the IC$_{50}$ doses of siRac1 NPs, IC$_{50}$ dose of DDP NPs and the concentration of siRca1 or DDP in siRNA/DDP NPs that provided the same effect, were examined in MDA-MB-231 cells, respectively (Fig. 7h). Then, the combination index was generated according to previous study[54]. The combination index was 0.69, which was less than one, indicating a synergistically inhibitory effect on tumor cell growth.

We next examined their pharmacokinetics (PK) and biodistribution (BioD) of siRac1/DDP NPs. PK was examined by intravenous injection of naked Cy5-siRac1, Cy5-siRac1/DDP NPs, Cy5-siRac1 NPs and DDP NPs to healthy mice (1 nmol siRNA dose per mouse, $n = 3$), respectively. With the protection of PEG outer layer[55,56], the NPs showed long blood circulation with a half-life ($t_{1/2}$) of around 4 h (Fig. 8a). In contrast, the naked siRNA was rapidly cleared from the blood (Fig. 8a). Then the PDXs from NAC resistant TNBC patients were used to evaluate the BioD via an intravenous injection of the Cy5-siRac1/DDP NPs into the PDX-bearing mice (Supplementary Table 7). The

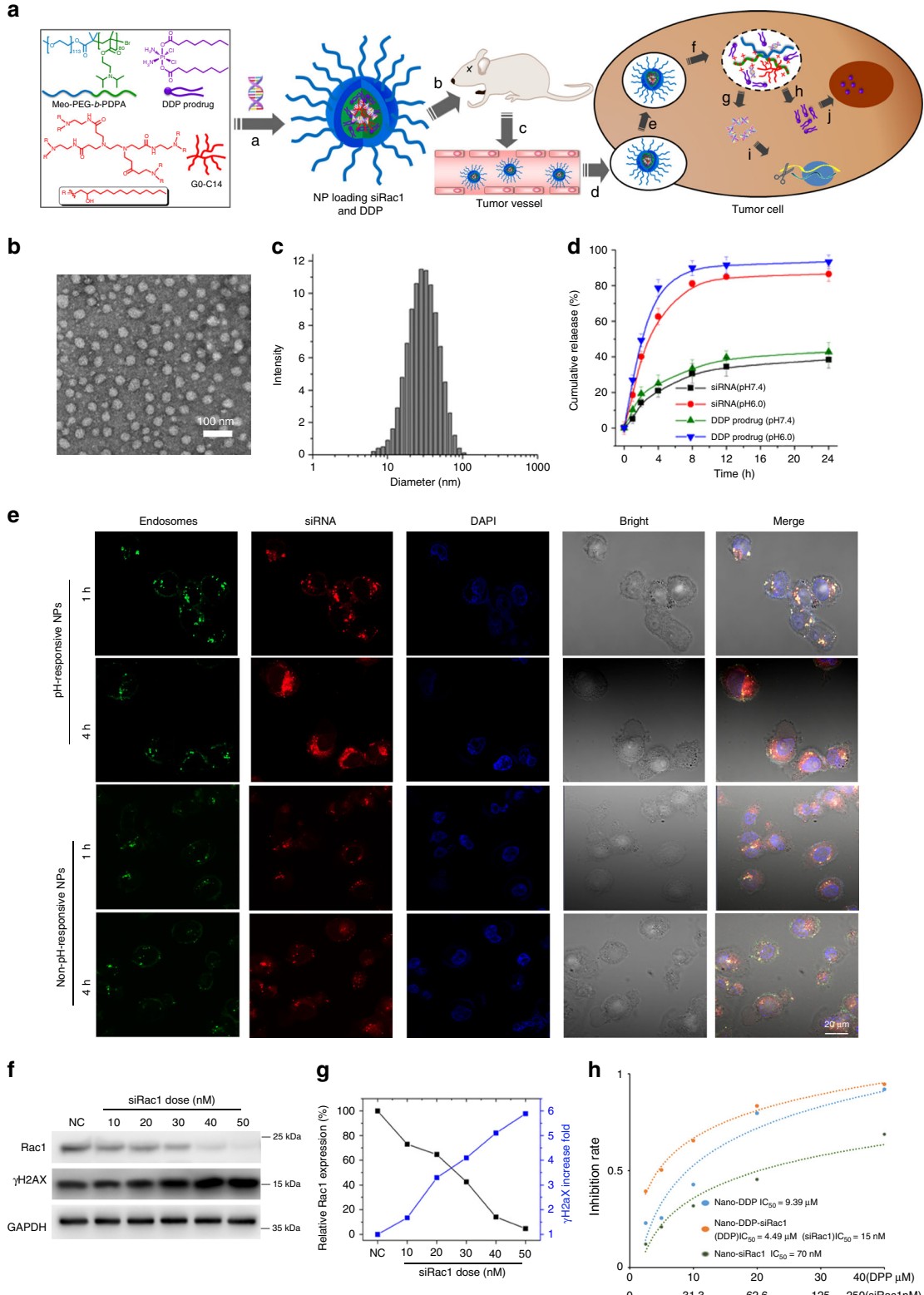

tumors and major organs were harvested for BioD quantification 24 h post injection. The Cy5-siRac1/DDP NPs showed a much higher tumor accumulation than the naked siRNA. The BioD of siRac1/DDP NPs demonstrated an ~7-fold higher siRNA accumulation in tumors than the naked siRNA (Fig. 8b and Supplementary Fig. 7A).

Then, we evaluated whether the siRac1/DDP NPs could silence Rac1 expression and recovered the chemosensitivity of NAC resistant breast tumors in the PDX model. The siRac1/DDP NPs were intravenously injected into the PDX-bearing mice once every two days at a 1 nmol siRNA dose per mouse ($n = 6$) (Supplementary Table 7). After three consecutive injections, the tumor growth was significantly inhibited compared to the mice treated with PBS (Control), blank NPs, or the NPs only loaded with siRac1 (Fig. 8c–e). The tumor size (Fig. 8d) and tumor weight (Fig. 8e) were about 10-fold larger in the NPs PBS

**Fig. 7 Synthesis and Characterization of siRac1 and PPD loaded nanoparticles (NP). a** Molecular structures of the pH-responsive polymer Meo-PEG-*b*-PDPA, DDP prodrug, and amphiphilic cationic lipid G0-C14 and schematic illustration of the endosomal pH-responsive NP platform for systemic delivery of siRac1 and DDP for synergistic breast cancer therapy. In aqueous solution, the Meo-PEG-*b*-PDPA polymer, DDP prodrug, and cationic lipid G0-C14 can co-assemble with siRNA to form stable NPs (**a**). After intravenous injection to mice (**b, c**), the NPs can extravasate from leaky tumor vasculature (**d**) and be internalized by the tumor cells (**e**). After cellular uptake, the endosomal pH-responsive characteristic of the Meo-PEG-*b*-PDPA polymer induces fast disassembly of the NPs (**f**), leading to the efficient endosomal escape and fast cytosolic release of siRNA (**g**) and DDP prodrug (**h**), which can respectively silence Rac1 expression (**i**) and induce DNA damage (**j**) to achieve synergistic breast cancer therapy. **b, c** TEM image (**b**) and size distribution (**c**) of the siRac1/DDP NPs in pH 7.4 PBS solution. The result was obtained over three independent experiments. **d** Cumulative siRac1 and DDP prodrug release from the Cy5-siRac1/DDP NPs incubated in PBS solution at a pH of 7.4 or 6.0. Bar graphs represent the mean ± SD of three independent experiments. **e** Cy5-siRac1/DDP escaped from the endosomes after they were delivered into tumor cell by pH-responsive NPs. CLSM images of the MDA-MB-231 cells incubated with the Cy5-siRac1/DDP NPs (pH-responsive or non-pH-responsive NPs) for 1 h and 4 h. The endosomes and nuclei were stained with Lysotracker green and Hoechst 33342, respectively. The result was obtained over three independent experiments. Scale bar, 20 μM. **f, g** The expression of Rac1 and γH2AX in the MDA-MB-231 cells treated with the siRac1/DDP NPs at different siRNA doses. The NPs loading with scrambled siRNA and DDP prodrug were used as negative control (NC). The result was obtained over three independent experiments. **h** The IC$_{50}$ of siRNA NPs or DDP NPs, the concentration of siRac1, C(siRac1), and the concentration of DDP, C(DDP), in siRNA/DDP NPs that provide the same effect, were examined MDA-MB-231 cells. Source data are provided as a Source Data file.

(Control) and blank NPs group than those in the siRac1/DDP NP group. The NPs loaded with siRac1 only or DDP prodrug only showed moderate efficacy to inhibit tumor growth (Fig. 8c–e), whereas the concurrent delivery of siRac1 and DDP efficiently inhibited tumor growth with a synergistic effect. The results of IHC staining further confirmed that NPs plus siRac1 significantly reduced Rac1 and p-ERK levels in the treated tumors (Fig. 8f, g). Consistently, the siRac1/DDP NPs are most effective in inducing DNA damage and reducing cell proliferation while inducing cell apoptosis, as measured by γH2AX, Ki67, and cleaved Caspase-3 staining (Fig. 8h–j).

To evaluate the systematic side effects of siRac1, several non-cancerous cell lines were used. Knocking down Rac1 did not affect the viability of MCF-10A (normalized human mammary epithelial cell), HEK-293 (human embryonic kidney cell), MCR-5 (human lung fibroblast) cells (Supplementary Fig. 7B). As the Rac1 siRNA we used also targeted murine Rac1, we evaluated whether siRac1/DDP NPs induced obvious systematic side effects. Noteworthy, multiple hematological parameters including aspartate aminotransferase, alanine aminotransferase, albumin, alkaline phosphatase, creatinine, and total protein range were in the normal range at 24 h post administration of siRac1/DDP NPs (Supplementary Fig. 7C). In addition, healthy mice also received the injection of siRac1/DDP NPs (1 nmol siRNA dose per mouse, *n* = 3). After three daily injections, no noticeable histological toxicity was detected in the tissues from heart, liver, spleen, lung, and kidney (Supplementary Fig. 7D). In the PDX-bearing mice, the siRac1/DDP NPs showed no obvious adverse influence on the mouse body weight when the treatment was completed 36 days post first injection (Supplementary Fig. 7E). These results indicate the good biocompatibility of this NP platform. Together, our data suggest that applying RNAi NP to carry siRac1 is an effective strategy to sensitize breast cancers to chemotherapy and to reverse the chemoresistance of breast cancers.

## Discussion

Inflicting DNA damage on cancer cells is one of the key anti-cancer effects of most chemotherapeutic agents. Therefore, enhanced repairing of the drug-induced DNA damage can confer resistance to chemotherapy. For instance, activation of nucleotide excision repair and homologous recombination mechanisms cause resistance to platinum-based drugs[57,58]. Moreover, nucleotide metabolism is critical for the DNA damage response. Bester et al. showed that nucleotide depletion in keratinocytes led to the replication fork stalling with ensuing DNA damage, which could be rescued upon nucleoside repletion[59]. Our group previously revealed

that PI3K promotes the Rac1 mediated cytoskeleton remodeling to activate aldolase by release this enzyme from cytoskeleton, thus enhances glycolytic flux and non-oxidative PPP[28,29]. Recent studies have shown that Rac1 activation promotes cancer progression via the PAK/RAF/ERK pathway[12,31]. In this study, we provided evidence that overexpression of Rac1 activated non-oxidative PPP and enhanced the nucleoside metabolism via activating aldolase A and ERK signaling separately (Fig. 6i). Consistently, we showed the prominent role of Rac1 in upregulating the R5P synthesis and nucleoside metabolism, thus promoting the repair of DNA damage caused by chemotherapy agents, and inducing chemoresistance of breast cancer cells to these drugs.

The somatic *Rac1* mutation P29S has been discovered as an oncogenic driver in the melanoma and other cancers[10–12]. High expression of Rac1 was shown to be associated with poor outcome in several human cancers, such as breast, colorectal cancers, and leukemia[13–16]. Our discovery of the pivotal role of Rac1 in enhancing nucleotide metabolism and inducing chemoresistance in multiple human cancers makes Rac1 as an attractive therapeutic target for potent sensitization to DNA damaging chemotherapies. Together with our studies, the potential contribution of Rac1 to the resistance to anti-cancer chemotherapeutic drugs, highlights the critical need to develop treatment strategies to target Rac1 related pathways in a clinical setting[12]. Several chemical compound were developed for targeting Rac1 and showed anti-cancer effects in cell lines or animal models[60,61]. However, the GTPases are still hard to be targeted clinically in spite of numerous efforts to develop their inhibitors. Herein, we applied siRNA to target Rac1 mRNA instead of its protein for cancer treatment. We developed an endosomal pH-responsive nanoparticle to carry the Rac1 siRNAs together with cisplatin, which resulted in an effective delivery of the Rac1 targeting oligonucleotide and cisplatin in breast tumors and exhibited a promising synergetic anti-tumor effect. Preclinical and clinical investigation has documented the successful delivery of therapeutic siRNA by nanoparticles to treat transthyretin (TTR) mediated amyloidosis (NCT01960348) and cancers[41]. Our current study provides a proof-of-principle that the sensitivity of chemotherapy drugs can be significantly increased by targeting Rac1 with endosomal pH-responsive nanoparticle encapsulated siRNAs.

Together, our data reveal the unacknowledged role of Rac1 in multiple chemoresistance of breast cancers by promoting the glycolysis in particularly non-oxidative PPP and nucleoside metabolism. Monitoring Rac1 level and targeting Rac1 may be utilized to predict and reverse the chemoresistance of breast cancers. Applying the pH-responsive nanoparticle that co-encapsulate Rac1 siRNA and cisplatin provided us a promising translational strategy to sensitize breast cancer to chemotherapies.

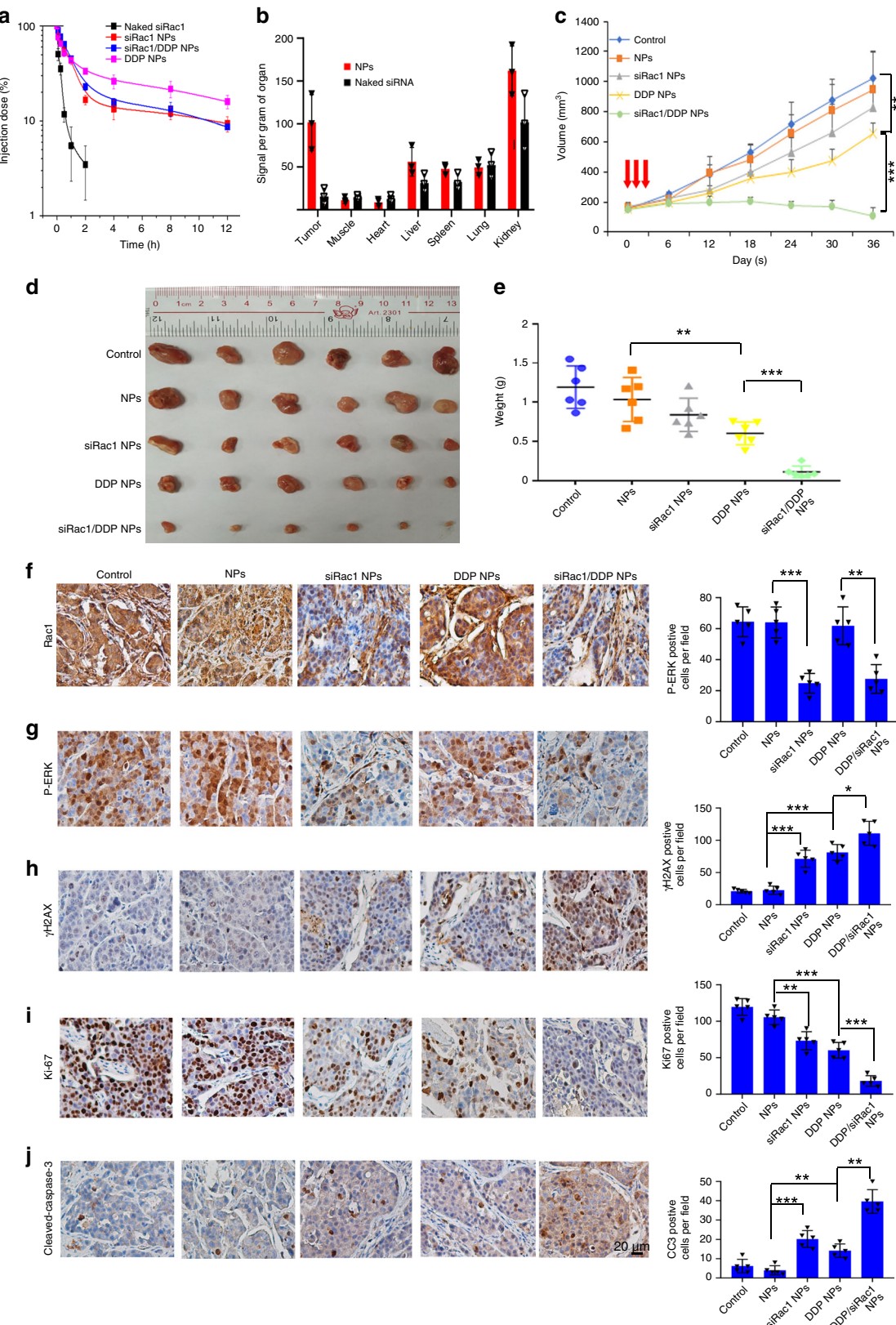

## Methods

**Patients and tissue samples**. Breast cancer samples were obtained from Breast Tumor Center, Sun Yat-Sen Memorial Hospital, Sun Yat-Sen University. In all, 198 surgical resected tumors and 133 core needle biopsies of breast cancer before neoadjuvant chemotherapy were collected from January 2010 to June 2013 and 2014 to 2017, respectively. The patients were followed up for 1–109 months and

7–37 months (median follow-up are 61 months and 43 months). Among the 198 tumor samples, there are 84 corresponding adjacent normal tissues. Pathological diagnosis, as well as ER, PR, and HER2 status, were verified by two pathologists independently. All human samples were collected with informed consents from the donors according to the International Ethical Guidelines for Biomedical Research Involving Human Subjects (CIOMS). The study was performed after

**Fig. 8 siRac1/DDP NPs recovers the chemosensitivity of PDXs derived from NAC resistant TNBC patients. a** Cy5-siRac1/DDP NPs showed much longer blood circulation half-life than the naked Cy5-siRac1. Bar graphs represent the mean ± SD of three independent experiments. **b** Cy5-siRac1/DDP NPs specifically concentrated in the breast tumors compared with the naked siRac1. The mice were killed at 24 h post injection of the naked Cy5-siRac1 and Cy5-siRac1/DDP NPs and then the biodistribution of the Cy5-siRac1 in the tumors and major organs of the PDX-bearing mice were detected. Bar graphs represent the mean ± SD of three independent experiments. **c–e** Tumor growth curves (**c**), tumor images (**d**), and tumor weights (**e**) of the PDXs from NAC resistant TNBC patients after systemic treatment with PBS, blank NPs, NPs loading siRac1 (siRac1 NPs), NPs loading DDP prodrug (DDP NPs), and siRac1/DDP NPs. Starting points of intravenous injections are indicated by the arrows. Tumors ($n = 6$ per group) were harvested at indicated days post injection. (Growth curve: control vs DDP NPs $p < 0.0001$, DDP NPs vs siRac1/DDP NPs $p < 0.0001$. two-way ANOVA + Dunnett's post hoc tests. Tumor weight NPs vs DDP NPs $p = 0.0006$, DDP NPs vs siRac1/DDP NPs $p < 0.0001$, two-sided unpaired $t$-test). **f–j** Immunohistochemistry analysis of the PDX tissues after systemic treatment in each group. Rac1 (**f**), p-ERK (**g**) γH2AX (**h**), Ki67 (**i**), CC3 (**j**). Scale bar, 20 μM. Each group $n = 5$. (**g**: NPs vs siRac1 NPs $p < 0.0001$, DDP NPs vs siRac1/DDP NPs $p$ $p = 0.0011$, H: NPs vs siRac1 NPs $p < 0.0001$, NPs vs DDP NPs $p < 0.0001$,DDP NPs vs siRac1/DDP NPs $p = 0.0178$, **i**: NPs vs siRac1 NPs $p = 0.002$, NPs vs DDP NPs $p = 0.0001$, DDP NPs vs siRac1/DDP NPs $p < 0.0001$, **j**: NPs vs siRac1 NPs $p < 0.0001$, NPs vs DDP NPs $p = 0.0006$, DDP NPs vs siRac1/DDP NPs $p < 0.0001$, two-sided unpaired $t$-test). Error bars shows mean ± SD of five random fields. *$p < 0.05$, **$p < 0.01$, and ***$p < 0.001$. Source data are provided as a Source Data file.

approval by the institutional review board (IRB) of Sun Yat-Sen Memorial Hospital

**Patient-derived xenograft experiments**. To establish patient-derived xenografts, TNBC tumor specimens were collected from three patients who had platinum-based NAC treatment with PD at Sun Yat-Sen Memorial Hospital, Sun Yat-Sen University (Guangzhou, China) between 2017 and 2018. The clinical features of patients were provided in Supplementary Table 7. All three patients was informed consent and approved by Institutional Review Board (IRB) of Sun Yat-Sen Memorial Hospital, Sun Yat-Sen University. Detailed procedure is described briefly below[62]. six-week-old NSG female mice were anaesthetized by isoflurance. The tumors were minced into 1 mm³ sized fragments and imbedded directly into the mammary fat pads. Once the PDX of first generation reached diameter of 1 cm, they were harvested and then minced into 1 mm³ sized fragments and imbedded directly into the mammary fat pads to establish the second generation for treatments. The PDX from each patient was transplanted to two mice. Every group has six PDX-bearing mice from three patients for treatments.

**Primers for PPP pathway genes in human breast cancer cell**.
Rac1 5′-ATGTCCCTGGCTGCTTATTGC-3′
  5′-CCAGTGTGTGCCACTTTTTGG-3′
  G6PD 5′-CGAGGCCGTCACCAAGAAC-3′
  5′-GTAGTGGTCGATGCGGTAGA-3′
  Pgls 5′-GGAGCCTCGTCTCGATGCTA-3′
  5′-GAGAGAAGATGCGTCCAGT-3′
  Pgd 5′-ATGGCCCAAGCTGACATCG-3′
  5′-AAAGCCGTGGTCATTCATGTT-3′
  Rpia 5′-AGTGCTGGGAATTGGAAGTGG-3′
  5′-GGGAATACAGACGAGGTTCAGA-3′
  Rpe 5′-TAGACTCTGGGGCCGATTATC-3′
  5′-GTCCTGGCCTAGCTGCTTTC-3′
  Tkt 5′-TCCACACCATGCGCTACAAG-3′
  5′-CAAGTCGGAGCTGATCTTCCT-3′
  Taldo 5′-CTCACCCGTGAAGCGTCAG-3′
  5′-GTTGGTGGTAGCATCCTGGG-3′
  Tktl1 5′-ACAAGCAGTCAGATCCAGAGA-3′
  5′-TAGCTGGCCCTGTCGAAGTA-3′
  Tktl2 5′-GGGACATGCTGCTCCTATCC-3′
  5′-CGTCAACAAACGGCAATCGG-3′

**RNA preparation and microarray analysis**. Breast cancer samples sensitive or resistant to neoadjuvant chemotherapy were collected from Sun Yat-Sen Memorial Hospital, Sun Yat-Sen University. Sensitive is for Completed Reponse (CR) or Partial Response (PR) after neoadjuvant chemotherapy and resistance is for Stable Disease (SD) or Progress Disease (PD) after neoadjuvant chemotherapy. This study was approved by the institutional review board (IRB) of Sun Yat-Sen Memorial Hospital. Surgical-resected tumor sample were frozen in liquid nitrogen immediately and stored at –80 °C freezer until usage. Total RNA was extracted using Trizol reagent. Agilent Human lncRNA Microarray V6 ($4 \times 180$ K) was used to analyze the global profiling of human lncRNAs and protein-coding transcripts in these samples. The microarray contains 83,835 lncRNAs and 27,233 coding genes. The raw data of the microarray were extracted by Feature Extraction software and further quantile normalized and exhibited as log2 transform using the GeneSpring software. The intensity was used to generate the heatmap by MeV4.7[63].

**Online dataset**. The correlation of Rac1 expression in breast cancer with clinicopathological features and survival outcome of TCGA database was analyzed on the UALCAN website (http://ualcan.path.uab.edu/analysis.html).

**Cell cultures and treatment**. MDA-MB-231, MDA-MB-436, BT-549, BT-474, SKBR3, ZR751, MCF-7 breast cancer cells, MCF-10A breast epithelial cells, A549 lung adenocarcinoma cells, SKOV3 (ovarian cancer) cells, AGS (gastric adenocarcinoma) cells, HEK-293 (human embryonic kidney cell), MCR-5 (human lung fibroblast) cells, mouse breast cancer cell 4T1 were obtained from American Type Culture Collection (ATCC) and grown according to standard protocols. MCF-10A cells were cultured in DMEM/F-12 with 5% horse serum, 20 ng/ml epidermal growth factor (EGF), 0.5 mg/ml hydrocortisone, 100 ng/ml cholera toxin, and 10 μg/ml insulin.

**Immunohistochemistry**. Immunohistochemistry (IHC) was performed according to the standard protocol. Briefly, the tissue sections were de-paraffinized by xylene, retrieved by 10% boiling sodium citrate, blocked with 10% goat serum, and incubated with primary antibody overnight at 4 °C. The following primary antibodies were used: Rac1 (Millipore, 1:800), γH2AX (Cell Signaling, 1:500), cleaved caspase-3 (Cell Signaling, 1:800), Ki67 (Cell Signaling, 1:300). The quantification of Rac1 expression was evaluated by two independent pathologists. Both sets of results were combined to give a mean score for further comparative evaluations. The IHC score, H-score, or Rac1 score, were determined by combining the percentage of positively stained tumor cells and the staining intensity of positively stained tumor cells[7]. The staining intensity was graded as follows: 0, no staining; 1, weak staining (light yellow); 2, moderate staining (yellow–brown); 3, strong staining (brown). The percentage of cells at each staining intensity level is calculated, and finally, an H-score is assigned using the following formula: [$1 \times$ (% cells 1+) + $2 \times$ (% cells 2+) + $3 \times$ (% cells 3+)].

Such method was used to evaluate Rac1 expression in breast cancer and adjacent normal samples. We use the median value as the cut-off to define Rac1-high and Rac1-Low in breast cancer samples for Rac1 expression analysis and that in all patients in neoadjuvant therapy evaluation.

**Cell migration and invasion assay**. Migration and invasion assays were carried out using 24-well Boyden chambers (Corning, USA) with 8M-inserts coated with fibronectin (Roche, USA) and Matrigel (BD, USA). One thousand MB-MDA-231 cells were seeded on the upper chamber without serum. Cells on the bottom of the upper chamber were counted 24 h after seeding.

**Flow cytometry**. Cell apoptosis was analyzed by flow cytometry. Cells were centrifuged at 1000 rpm for 5 min and washed with cold PBS twice. Annexin IV (20 μg/ml final concentration) and Propidium Iodide staining solution (50 μg/ml final concentration) were added to the cells and incubated for 30 min at 37 °C in the dark. Ten thousand cells were analyzed using a CytomicsTM FC 500 instrument (Beckman Coulter, USA) equipped with CXP software. Data was processed by Flowjo V10. In the apoptosis analysis, gating of the flow cytometry data was according to the cells in siCTL or Vector group without Annexin V and Propidium Iodide staining, as indicated in the Supplementary Fig. 5J–L.

**Immunoblotting**. Cells were lysed in RIPA lysis buffer (Beotime, China) supplemented with protease and phosphatase inhibitors (Life Technologies, USA). Protein samples were subjected to 8–10% SDS-PAGE according to the mass of protein and transferred to PVDF membranes (Bio-Rad, USA). Membranes were then blocked with 5% non-fat milk in 0.1% TBST buffer for 1 h and incubated with primary antibody overnight at 4 °C. Primary antibodies against phospho-PAK1/2 (Cell Signaling, 1:300), total PAK1 (abcam, 1:500), phospho-C-Raf (Cell Signaling, 1:1000), total C-Raf (Cell Signaling, 1:1000), phospho-Mek1/2 (Cell Signaling, 1:1000), total Mek1/2 (Cell Signaling, 1:1000), phospho-Akt (S473) (Cell Signaling, 1:1000), total Akt (Cell Signaling, 1:1000), Rac1(Millipore,1:1000), total Erk (Cell Signaling, 1:1000), phospho-Erk (Cell Signaling,1:1000), γH2AX (Cell Signaling,

1:1000), Aldolase A (Cell Signaling, 1:1000), and GAPDH (Cell Signaling, 1:10,000) were used. Standard procedures were used for immunoblotting.

**Colony formation assay**. One thousand cells were plated in 6-well plates and cultured for 10 days. The colonies were stained with 1% crystal violet for after fixation with 4% formaldehyde for 15 min.

**MTS cell viability assay**. One thousand cells were seeded each well in 96-well plates. At each time point, cells were stained with sterile MTS mix liquid (1:10 in culture median) for 2 h at 37 °C in the dark. The absorbance was measured at 492 nm.

**Drugs**. DDP (P3494) and doxorubicin (D1515) were purchased from Sigma. Docetaxel (T1034) was purchased from TargetMol. Carboplatin (C805203) was purchased from Macklin, ERK inhibitor SCH772984 and Rac1 inhibitor NCS23766 was from Selleck.

The DDP prodrug was synthesized according previous report[51], by using the reaction between *cis, trans, cis*-[PtCl$_2$(OH)$_2$(NH$_3$)$_2$], and sebacic anhydride. The structure of this DDP prodrug was analyzed by proton nuclear magnetic resonance (HNMR) to confirm successful synthesis (Supplementary Fig. 8). Details of the drugs including their targets and concentrations noted in the figure legends.

**siRNA /shRNA and constructs**. Two siRNA/shRNA duplexes that target Rac1 were used. siRNA1/shRNA1 (5′-AGACGGAGCTGTAGGTAAA-3′) targets the coding region of homo sapiens Rac1. siRNA2/shRNA2 (5′-CCTTTGTACGCTTTGCTCA-3′) targets the 3′ UTR of homo sapiens Rac1 (NM_006908.5). Coincidently, the siRNA2 also targets the 3′ UTR of mus musculus Rac1 (NM_001347530.1). Therefore, siRNA2 was also used to silence mouse Rac1.

Two siRNA duplexes that target PAK1 were used. Both siRNA1 (5′-GCCTAGAC ATTCAAGACAA-3′) and siRNA2 (5′-CAAAGATGCTGGAACCCTA-3′) targets the coding region of PAK1.

The control siRNA/shRNA sequence is as follows: 5′-UAAGGCUAUGAAGA GAUAC-3′. The plko-tet-on "all-in-one" plasmid was used to generate the inducible expression of shRac1 and shControl.

Aldolase A (WT, D33S, and R42A), ERK1 and GFP-Rac1 in PCDNA3.1 plasmid were used for ectopic expression in breast cancer cells. Lipofectamine 3000 (Invitrogen) was used for the siRNA or PCDNA3.1 plasmid transfection.

**Immunofluorescence**. Cells were treated with ionizing radiation (IR) 2 Gy for indicated time periods after transiently transfected with siCTL, siRac1-1 and siRac1-2 for 48 h. In brief, Cells were fixed by 4% formaldehyde and permeated by 0.1% Triton 100X in PBS for 15 min, blocked with 10% goat serum for 1 h, and incubated with primary antibody overnight at 4 °C. After being washed with PBS, the Cells were incubated with fluorescent-labeled secondary antibody. Cells were stained against γH2AX (1:50, Cell signaling) and examined with a ZeissAxiovert 200 M fluorescence microscope.

**Dual luciferase reporter assay**. RPIA and Tkt promotors were separately sub-cloned into a pGL4.17[luc2/Neo] vector which contains the firefly luciferase gene (Promega, USA) to establish two constructs, RPIA-luc and Tkt-luc, respectively. The pGL4.17[luc2/Neo] vector containing the Renilla luciferase gene acted as an internal control. Vector, RPIA-luc and Tkt-luc were transfected with into MDA-MB-231, respectively, then MDA-MB-231 cells were treated with or without ERK inhibitor SCH772984. Luciferase activities were detected overnight after transfection by the Dual-Luciferase Reporter Assay System (Promega, Madison, WI) according to the manufacturer's instructions.

**Metabolism analysis**. For steady-state studies, metabolite fractions were resuspended in high-performance liquid chromatography (HPLC)-grade water and analyzed by targeted LC-MS/MS using a 5500 QTRAP mass spectrometer (AB/SCIEX) coupled to a Prominence UFLC HPLC system (Shimadzu) with Amide HILIC chromatography (Waters). Data were acquired in selected reaction monitoring (SRM) mode using positive/negative polarity switching for steady-state polar profiling. Peak areas from the total ion current for each metabolite SRM transition were integrated using MultiQuantv2.0 software (AB/SCIEX).

**Glucose uptake assays**. SiRac1 MDA-MB-231 or exogenous GFP-Rac1expression MDA-MB-436 cells were cultured to ~50% confluence in continuous log phase and. In growth media on 6 cm dishes. A complete media change was performed three hours prior to metabolite collection. The cells were treated with 1 mM 13 C labeled 2DG for 30 seconds and metabolites collected with 70% methanol extraction for LC-MS/MS. Alternatively, glucose uptake was measured with a kit from abcam (ab136955)

**Cell permeabilization, fractionation, and determination of aldolase A levels**. For permeabilization, same number of cells cultured in 6-well dishes were washed with PBS and then incubated in 30 μg/ml digitonin/PBS for 5 min at 4 °C. After incubation, the supernatant was collected and the pellet was lysed with 200 μl of lysis buffer for each well. The supernatant was centrifuged at 2000 rpm to remove cellular components. In total, 40 μl of supernatant (8% of total supernatant) of 20 μl of cell lysate (10% of total lysate) were run on the same SDS PAGE, transferred to PVDF membrane for immunoblotting[28].

**Aldolase enzymatic assay**. The aldolase enzymatic assay was performed based on Boyer's modification of the hydrazine assay[64] in which 3-phosphoglyceraldehyde reacts with hydrazine to form a hydrazone which absorbs at 240 nm. Before performing aldolase enzymatic assay on cells grown in 6-well plates, the cell supernatant or cell lysate was mixed with 3 μl EDTA (0.01 M), 6 μl iodoacetate (0.01 M), 200 μl hydrazine (0.0035 M), and appropriate volume of lysis buffer to make a final volume of 300 μl. A blank read was taken at 240 nm, 10 μl of 0.12 M FBP were added and absorption was detected at 240 nm in 5 min intervals for three readings. Mean enzymatic activity was determined according to: {[A(15 min)−A(10 min)] +[A(10 min)−A(5 min)]+[A(5 min)−A(0 min)]}/3.

For the detection of aldolase activity in whole cells, cells were grown in UV-transparent 96-well plates, treated for three hours, and lysed in 30 μl Digitonin solution (100 μg/ml PBS), then 60 μl of the EDTA/Iodoacetate/hydrazine solution were added followed by addition of 2.5 μl FBP and absorption was read in a BioTek plate reader at 240 nm. Enzymatic activity was determined in triplicate, normalized to the control, and means and standard deviations calculated.

**14C-labeling experiment**. Cells grown to 60–70% confluence in a 10-cm tissue culture dish were treated with 6-14C- or 1-14C-glucose for 8 h. DNA was extracted using Qiagen DNeasy kit (catalog no. 69504) according to the manufacturer's instructions. Equal amount of DNA were added to scintillation vials, and radio-activity was measured by liquid scintillation counting and normalized to the DNA concentration. All experiments were done in triplicates.

**Animal experiments**. Female Balb/c nude mice (athymic nude (nu/nu)) and NSG (NOD/SCID/IL2Rγ$^{null}$) mice of 4–6-week-old were purchased from Beijing VitalRiver Laboratory Animal Technology Co., Ltd. All animal work was conducted in accordance with a protocol approved by the Institutional Animal Care and Use Committee at the Medical School of Sun Yat-Sen University and laboratory animal facility has been accredited by AAALAC (Association for Assessment and Accreditation of Laboratory Animal Care International) and the IACUC (Institutional Animal Care and Use Committee) of Guangdong Laboratory Animal. Monitoring Institute approved all animal protocols used in this study. Mice were bred in specific pathogen free (SPF) animal house with 28 °C and 50% humidity. Indicated cells were inoculated into mammary pad of the six-week-old female nude mouse (*n* = 5 per group). To provide estrogen for MCF-7 and MCF-7R tumor growth, each mouse was implanted with a 1.7 mg 17β-estradiol pellet (60-day release, Innovative Research of America, Sarasota, FL, USA) 3 days before inoculation of MCF-7 and MCF-7R cells. After the xenografts became palpable (~150 mm$^3$), mice were injected with DDP (4 mg/kg weekly) or doxorubicin (2 mg/ kg weekly) intraperitoneal or fed with doxycycline (0.5 mg/ml) in drinking water with 2% sucrose for 3 weeks.

For the nanodrug experiments, MDA-MB-231 tumor-bearing NSG female mice were randomly divided into five groups (*n* = 5). After the xenografts became palpable (around 150mm$^3$), mouse was intravenously injected with (i) PBS, (ii) blank NPs, (iii) siRac1 NPs, (iv) DDP NPs, or (v) siRac1/DDP NPs at a 1 nmol siRNA dose per mouse once every two days. All the mice were administrated by administered three consecutive injections were injected with indicated dose through tail vein. The animals were killed when the xenografts reached at ~1500 mm$^3$. Tumors and harvested organs were subjected to H&E and further IHC staining.

**Preparation of Nanoparticles**. The siRac1 NPs, DDP NPs, or siRac1/DDP NPs were prepared according to our previous report[47]. Meo-PEG-*b*-PDPA polymer was dissolved in N,N′-dimethylformamide (DMF) to form a homogenous solution with a concentration of 10 mg/mL. Subsequently, a mixture of 1 nmol siRNA2 of Rac1 (0.1 nmol/μL aqueous solution), 50 μL of G0-C14 (5 mg/mL in DMF), and 10 μL of DDP prodrug (10 mg/mL in DMF) was prepared and then mixed with 200 μL of Meo-PEG-*b*-PDPA solution. For siRac1 NPs, a mixture of 1 nmol siRNA2 of Rac1 (0.1 nmol/μL aqueous solution), 50 μL of G0-C14 (5 mg/mL in DMF) was prepared and then mixed with 200 μL of Meo-PEG-*b*-PDPA solution. For DDP NPs, a mixture of 10 μL of DDP prodrug (10 mg/mL in DMF), 50 μL of G0-C14 (5 mg/mL in DMF) was prepared and then mixed with 200 μL of Meo-PEG-*b*-PDPA solution. Under vigorously stirring (1000 rpm), the mixture was added dropwise to 5 mL of deionized water. The NP dispersion formed was transferred to an ultrafiltration device (EMD Millipore, MWCO 100 K) and centrifuged to remove the organic solvent and free compounds. After washing with PBS buffer (pH 7.4) (3 × 5 mL), the drug loaded NPs were dispersed in 1 mL of PBS buffer (pH 7.4).

For preparation of non-pH-responsive NPs, the commercially available polymer, Meo-PEG-b-PLGA was used. The non-pH-responsive NPs was then used to load the siRNA/DDP as described above.

**Characterizations of NPs**. Size and zeta potential were determined by dynamic light scattering (DLS, Brookhaven Instruments Corporation). The morphology of NPs was visualized on a Tecnai G2 Spirit BioTWIN transmission electron

microscope (TEM). Before observation, the sample was stained with 1% uranyl acetate and dried under air. To determine siRNA encapsulation efficiency (EE%), Cy5-labeled siRac1 (Cy5-siRNA-DDP) loaded NPs were prepared according to the method aforementioned. A small volume (5 μL) of the NP solution was withdrawn and mixed with 20-fold DMSO. The standard was prepared by mixing 5 μL of naked Cy5-siRNA solution (1nmol/mL in pH 7.4 PBS buffer) with 20-fold DMSO. The fluorescence intensity of Cy5-siRNA-DDP was measured using a microplate reader and the siRNA EE% (−80%) is calculated as: EE% = (FI$_{NPs}$/FI$_{Standard}$) × 100. The DDP prodrug EE% (~50%) was determined by atomic absorption spectroscopy.

**Evaluation of endosomal escape.** The Cy5-siRac1/DDP NPs were prepared according to the method described above and incubated with the MD-MBA-231 cells for 1 or 4 h. Subsequently, the medium was removed and washed with PBS thrice. After respectively staining the nuclei and endosomes with Hoechst 33342 and Lysotracker green, the cells were viewed under ZeissAxiovert 200 M fluorescence microscope.

**Pharmacokinetics study.** Healthy male BALB/c mice were randomly divided into two groups ($n = 3$) and given an intravenous injection of either naked cy5-siRNA or cy5-siRNA-DDP loaded NPs at a 1-nmol siRNA dose per mouse. At pre-determined time intervals, orbital vein blood (20 μL) was withdrawn using a tube containing heparin, and the wound was pressed for several seconds to stop the bleeding. The fluorescence intensity of cy5-labeled siRNA in the blood was determined by microplate reader.

**Combination index.** The combination index (CI) was calculated according to ref. [54]. Briefly, we calculated the IC$_{50}$ of siRac1 NPs (IC$_{50-siRac1-A}$), IC$_{50}$ of DDP NPs (IC$_{50-DDP-A}$) and the concentrations of siRac1and DDP B contained in siRac1/DDP NPs combination that provide the same effect, denoted as (C$_{siRca1}$, C$_{DPP}$), respectively. The CI was calculated as: CI = C$_{siRac1}$/ IC$_{50-siRac1-A}$ + C$_{DDP}$/IC$_{50-DDP-A}$. A CI of less than, equal to, and more than 1 indicates synergy, additivity, and antagonism, respectively.

**Biodistribution.** Tumor-bearing female Athymic nude mice were randomly divided into two groups ($n = 3$) and given an intravenous injection of either naked Cy5-siRNA or cy5-siRNA-loaded NPs at a 1-nmol siRNA dose per mouse. Twenty-four hours after the injection, the mice were imaged using the Maestro 2 In-Vivo Imaging System (Cri Inc). Organs and tumors were then harvested and imaged. To quantify the accumulation of NPs in tumors and organs, the tissues were homogenized and fluorescence intensity of the Cy5-siRac1 in each organ was examined by microplate reader.

**Histology and hematology.** Healthy female BALB/c mice were randomly divided into 5 groups ($n = 3$) and administered daily intravenous injections of either PBS, pure nanoparticle, Nano-siRac1, Nano-DDP or Nano-siRac1-DDP at a 1-nmol siRNA dose per mouse. After three consecutive injections, the main organs were collected 24 h post the final injection, fixed with 4% paraformaldehyde, and embedded in paraffin. Tissue sections were stained with hematoxylin-eosin (H&E) and then viewed under an optical microscope. Blood samples were collected for AST, ALT, ALP, Urea, CREA, GLB, and ALB examination.

**Statistics.** The in vitro data were presented as mean ± S.D. of three independent experiments. All statistical analyses were performed using SPSS 16.0 statistical software package and Graphpad Prism 8. Unpaired two-sided Student's $t$ test and one-way ANOVA was used to compare cell viability, colony formation, apoptosis and tumor volume with different treatments, and post hoc tests were used to test difference between groups. Chi-square test was used to analyze the relationship between RAC1 expression and clinicopathological status. Kaplan-Meier curves and log-rank test were used to compare overall survival (OS) and disease-free survival (DFS) in different patient groups. Wald test was used in multivariate Cox proportional hazard analysis of expression of Rac1 and disease-free survival (DFS). Kolmogorov–Smirnov test was used to calculate Enrichment score (ES) and permutation test was used to exam the $p$ value in Gene Set Enrichment Analysis of the mRNA expression profiles of the NAC TNBC. In all cases, $*p < 0.05$, $**p < 0.01$ and $***p < 0.001$.

## Data availability

The expression profile microarray data of breast cancer cells and tissues have been deposited in the ArrayExpress database under the accession code E-MTAB-8787. The microarray data referenced during the study are available in a public repository from the website (https://www.ebi.ac.uk/arrayexpress/experiments/E-MTAB-8787). The source data underlying Figs. 1a, b, 2a–l, 3a–n, 4a–g, 5a, c, 6c, d, f, h, and 7a–c, e–j, Supplementary Figs. 1a–b, f–k, 2a–h, 3a–h, k–l, 4a–g, 5a–e, g, 6a–e, h–j, and 7b, c, e are provided as a Source Data file. All the other data supporting the findings of this study are available within the article and its supplementary information files and from the corresponding author upon reasonable request. A reporting summary for this article is available as a Supplementary Information file.

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

## Acknowledgements

This work was supported by grants from the National Science and Technology Major Project(2020ZX09201021), the Natural Science Foundation of China (81672738, 81730077, 81572596, U1601223, and 81972471), the National Key Research and Development Program of China (2016YFC1302301), the Sun Yat-Sen University Clinical Research 5010 Program (2018007), the Sun Yat-Sen Clinical Research Cultivating Program (SYS-C-201801), Program from Guangdong Introducing Innovative and Entrepreneurial Teams (2016ZT06S252), Natural Science Foundation of Guangdong Province (2017A030313828),, Guangzhou Science and Technology Bureau (201704020095), the Fundamental Research Funds for the Central Universities (17ykjc14) Guangdong Science and Technology Department (2017B030314026).

## Author contributions

H.H., H.Y., and E.S. contributed to conceptualization; H.H., H.Y., E.S., T.Q., Q.L., Z.B., H.M., D.L., J.C., W.W., X.L., F.Z., Y.Y., M.L.L., P.E.S., X.X., and G.W. contributed to methodology; H.H., Q.L., Z.B., and H.M. contributed to validation; H.H., H.Y., T.Q., E.S., Q.L., Z.B., H.M., D.L., and J.C. contributed to formal analysis; H.H., H.Y., E.S., Q.L., Z.B., H.M., D.L., J.C., and W.W. contributed to investigation; H.H., H.Y., E.S., and W.F. contributed to resources; H.H., H.Y., E.S., Q.L., T.Q., and Z.B. contributed to data curation; H.H., P.E.S., X.X., and G.W. contributed to article writing.

## Competing interests
The authors declare no competing interests.
