## [Peer Review File · Nature Communications]

Reviewers' comments:

Reviewer #1 (Remarks to the Author); expert in GTPases/Rac1 signalling:

In this ms, the authors examine the relationship of Rac1 expression to chemotherapy resistance in breast cancer and delineate a mechanism that might explain these effects. Specifically, they point to Rac1-induced membrane release of aldolase, plus ERK-mediated effects on the non-oxidative PPP, as a means of regulating nucleotide levels and thereby altering sensitivity to DNA damaging agents such as cisplatin. They further demonstrate a potential clinical application of their findings, but encapsulating a Rac1 siRNA plus cisplatin in a nano-particle which they deliver to xenografted mice.

the work is in general well-done and the topic is timely and interesting. However, the work falls short in a number of areas.

1) The work is poorly written. Normally, I wouldn't make this my first point, but the grammar, usage, and spelling mistakes are so frequent and glaring that it distracts from reading the ms. This work needs to be carefully reviewed by a native speaker.

2) While the authors are enamored of their RNAi delivery method, they give short shrift to the existing preclinical small molecule Rac1 inhibitors. In any event, one or more of these should be used to augment the siRNA data in at least one of the early figures.

3) The immunoblot data in Fig. 3C and D need to be better quantified. As shown, I can't see much increase in total Rac1 levels (3D) or difference in H2AX staining (3E, lanes 5,6).

Reviewer #3 (Remarks to the Author); expert in triple-negative breast cancer:

In this manuscript, Li et al demonstrated the involvement of Rac1 in chemo-resistance in breast cancer. They identified Rac1 was upregulated in the resistant breast cancer from RNA-seq analysis. Through in vitro and in vivo studies, they showed that knockdown of Rac1 could make MDA-MB-231 triple negative breast cancer cells more chemosensitive by inducing DNA damage via glycometabolism and non-oxidative pentose phosphate pathway. In addition, they also found that knockdown of Rac1 could re-sensitize MCF-7DR to doxorubicin (DOX). They demonstrated that delivery of nanoparticle (NP containing siRNA against Rac1 and cisplatin (DDP) could be a novel strategy for breast cancer treatment by patient derived xenograft model established from triple negative breast cancer. However, results in the current format is not strong enough to support their claim. The study needs to be extensively revised.

Major concern

1. Line 102 states, among the 16 common overexpressed genes, Rac1 overexpression was particularly noted, because Rac1 was not only expressed higher in breast tumor samples than normal tissues (Figure S1C), but also its overexpression correlated with advanced tumor stage. Firstly, it is difficult to believe there is significant statistical difference on Fig S1C. The error bar for tumor is so large, and it appears at a very similar transcript level as that of normal. The same applies for Fig S1D.

2. In line 133, to further validate the expression of Rac1 in chemo-resistant breast cancers, immunohistochemistry was applied in 198 cases of breast cancer tissues (Table S2). There is no indication that these 198 cases are in fact chemo-resistant breast cancers. They seem to be merely a cohort of breast cancer cases of which there are at least 69 cases which are triple negative (calculations based on data given in Table S2. Not all would have necessarily received chemotherapy, given that 61% of these cases are estrogen receptor positive. Also in line 119-120,

it says "when analyzing the correlation of Rac1 expression with different subtypes of breast cancer, we discovered that the TNBC patients with high Rac1 levels had worse OS and DFS (Fig 1 H-I). However the number of cases in these 2 graphs total to 101 which is more than what should be expected from Table S2.

3. Table S4 gives multivariate Cox proportional hazard analysis of 198 cases of patients with breast cancer, whilst Table S5, that of the 101 cases of TNBC patients. Line 134 – 138 however quote the data of tables S4 and S5 as suggesting that both this data are supportive that high Rac1 level is associated with chemo-resistance of breast cancers. Only the 101 TNBC patients had received neoadjuvant chemotherapy. There is no indication what sort of treatment the 198 cases had received, hence it is incorrect to use this data in support of chemo-resistance.

4. Based on their hypothesis, they proposed that high expression level of Rac1 contribute to resistance chemotherapy in particular DOX and DDP. It would be expected that the authors would determine the effect of Rac1 overexpression on the sensitivity to DDP or DOX. Instead, most of the study is based on Rac1 knockdown approach, with only one experiment rather demonstrating effect of overexpression, and as supplemental data (Figure S3 H, I). The results from knockdown experiments should be supportive evidence, rather than direct evidence for the hypothesis.

5. In line 143, they chose MDA-MB-231 based on the expression of Rac1 (Figure S2A). MDA-MB-231 is a triple negative breast cancer cell only. This does not mean the line is resistant to DDP. In line with the study, they should first determine the sensitivity of different breast cancer cell lines to DDP by IC50 study, and show the expression of Rac1 in these cell lines. Interestingly, in Figure S2A, MCF-7 which is chemosensitive shows almost as high Rac1 expression as that of MDA-MB-231.

Once they have confirmed the Rac1 expression level and IC50 to DDP, they would be able to determine the effect of Rac1 overexpression on DDP sensitivity in the low Rac1 expression cells, and conversely, determine the effect of Rac1 knockdown on DDP sensitivity in the high Rac1 expression cells.

6. Line 163 states the effect of Rac1 knockdown on the IC50 of DOX in MCF-7DR. However the legend of Fig S3 (A-C) states IC50 of DDP on NCF-7DR cells. Is it DOX or DDP? It would seem like it should be DOX. Fig S3 (H,I) compares the IC50 of DDP in MDA-MB-231 cells which overexpressed with Rac1. It would be better if they showed the effect on IC50 of DDP in MDA-MB-231 for both overexpression and knockdown, since the majority of the study is based on the use of DDP. Moreover, the labeling of the subpanels of Fig S3 are largely incorrect.

7. In Figure 3, they determined the amount of different metabolites by mass spectrometry and identified that knockdown of Rac1 would reduce the level of various metabolites of glucose metabolism. Thus, they claimed that Rac1 could promote glycolysis and non-oxidative PPP (line 261). I agree that Rac1 can affect non-oxidative PPP thus affecting the nucleotides for DNA repair. However, their metabolites' measurement is insufficient support for the effect on glycolysis as they did not determine the rate of glucose utilization (glucose to glucose-6-phosphate, the first step in glycolysis) in their study. They merely mention their previous studies revealed that the activation of PI3K-Rac-1-cytoskeleton axis enhanced the glycolysis and non-oxidative PP through promoting aldolase A activity but do not cite any reference. Therefore, the claim is not sufficiently valid.

8. The effect of Rac1 in breast cancer on glycolysis and non-oxidative PPP was not clear. The mechanistic detail of Rac1 on metabolism was missing in the current study.

9. In figures 2 and 3, they mentioned that knockdown of Rac1 would induce DNA damage by determination of γ H2AX expression. To match their hypothesis, they should determine if overexpression of Rac1 will result in reduced DNA damage under DDP treatment.

10. In figure 6E, the subcellular localization of Rac1 siRNA seems most unusual as majority of the siRNA appear to be localized in the cell membrane.

11. In figures 7F to J, they showed bar charts but statistical analyses were missing.

12. The authors claim that knockdown of Rac1 could induce DNA damage (Figure 2 N-P) but these results are at most borderline. They also show the NP containing Rac1 siRNA and DDP (Figure 7B) showed no tissue specificity. Therefore, they authors should test the effect of Rac1 knockdown on cell viability of different non-cancerous cell lines to address if the treatment will have a potential to induce undesirable effect in normal human tissue. Furthermore, although they did perform mouse experiment using NP with the siRNA only, the siRNA is specific to human Rac1, therefore, the

mouse model cannot determine the undesirable effect in humans.

13. In line 468, the authors mentioned that Rac1 is important in different human cancers. The authors should discuss whether Rac1 overexpression will be able to induce resistance to DDP in other human cancers in addition to breast cancer. If possible, they could determine the expression of Rac1 in different drug resistant cell lines in order to provide evidence to support whether Rac1 may also affect DDP sensitivity in human cancers besides breast cancer. Since Rac1 is a small GTPase, the authors should determine if any mutation will lead to constitutive activated Rac1 in human cancers.

Minor concern

1. Line 31. The second word should be "Mechanistically" not mechanically.
2. MDA-MB-231 was written as MD-MBA-231 in few places (e.g lines 143, 144, 148..etc).
3. γ H2AX was written as γ H2ax (lines 180, 274).
4. The relation between Rac1/ERK/aldolase was not clearly described.
5. The authors should consider if Rac1 knockdown led to DNA damage or Rac1 knockdown compromised DNA repair. Although the outcome may be similar, but the concept is different.

The point-by-point response to all the questions is presented below.

Concern of Reviewer 1

1) The work is poorly written. Normally, I wouldn't make this my first point, but the grammar, usage, and spelling mistakes are so frequent and glaring that it distracts from reading the ms. This work needs to be carefully reviewed by a native speaker.

We are sorry for these mistakes. We have rewritten the manuscript carefully and improved its readability.

2) While the authors are enamored of their RNAi delivery method, they give short shrift to the existing preclinical small molecule Rac1 inhibitors. In any event, one or more of these should be used to augment the siRNA data in at least one of the early figures.

Following the reviewer's suggestion, Rac1 inhibitor NSC23766 has been used to treat the cells, including MDA-MB-231, A549CR (A549, lung cancer cells that are resistant to carboplatin), SKOV-3 (ovary cancer cell) and ASG (gastric cancer cell) cells. NSC23766 reduced the IC50 of DDP in MDA-MB-231 (Figure S3D), SKOV-3, ASG cells (Figure S3G, H) and IC50 of carboplatin in A549CR (Figure S3F), resembling the effect of Rac1 RNAi.

In addition, we also revealed that Rac1 inhibition with NSC23766 decreased p-ERK level in MDA-MB-231 cells (Figure S4C), similar as Rac1 RNAi in the cancer cells (Figure 3F).

3) The immunoblot data in Fig. 3C and D need to be better quantified. As shown, I can't see much increase in total Rac1 levels (3D) or difference in H2AX staining (3E, lanes 5,6).

In original Figure 3D, GFP-Rac1 was transfected into MDA-MB-231 cells. Thus, the endogenous Rac1 which was detected by anti-Rac1 would not increase. The GFP-Rac1 had been blotted by anti-GFP as a different band from the endogenous one. In the revised version, we replaced MDA-MB-231 by MDA-MB-436 cells following the reviewer's suggestion and got similar results (Figure 3E).

In addition, we reperformed the experiment and optimized the condition of the western blotting for Figure 3F (original Figure 3E), which showed more obvious difference of γ H2AX levels.

Major concern of Reviewer 3:

1. Line 102 states, among the 16 common overexpressed genes, Rac1 overexpression was particularly noted, because Rac1 was not only expressed higher in breast tumor samples than normal tissues (Figure S1C), but also its overexpression correlated with advanced tumor stage. Firstly, it is difficult to believe there is significant statistical

difference on Fig S1C. The error bar for tumor is so large, and it appears at a very similar transcript level as that of normal. The same applies for Fig S1D.

The Rac1 levels in Fig. S1C and S1D are downloaded from TCGA database. The P values were generated by the UALCAN website (<http://ualcan.path.uab.edu/analysis.html>) automatically, which were shown below. Following the reviewer’s suggestion, we have provided the exact P value in the related figures.

2. In line 133 (should be 113), to further validate the expression of Rac1 in chemo-resistant breast cancers, immunohistochemistry was applied in 198 cases of breast cancer tissues (Table S2). There is no indication that these 198 cases are in fact chemo-resistant breast cancers. They seem to be merely a cohort of breast cancer cases of which there are at least 69 cases which are triple negative (calculations based on data given in Table S2). Not all would have necessarily received chemotherapy, given that 61% of these cases are estrogen receptor positive. Also in line 119-120, it says “when analyzing the correlation of Rac1 expression with different subtypes of breast cancer, we discovered that the TNBC patients with high Rac1 levels had worse OS and DFS (Fig 1 H-I). However the number of cases in these 2 graphs total to 101 which is more than what should be expected from Table S2.

As the reviewer pointed out, the 198 tissues are merely a cohort of breast cancer

cases. We didn't mean that they were chemo-resistant breast cancers. We used this cohort to study the association of Rac1 with the outcome of all and different subtypes of breast cancer patient (Figure 1G-1H), as well as with clinicopathological parameters (Table S2). The 133 core needle biopsies of breast cancer before neoadjuvant chemotherapy are the cohort that we used to evaluate the association of Rac1 level with chemoresistance (Figure 1J-1L, Table S3). We have re-phased the sentence in line 113 to avoid misunderstanding.

However, we did make mistakes in Table S2 and S3. The numbers of positive and negative ER and PR patients were inverted. There are 121 ER negative cases, including 2 PR positive cases, 15 HER2 positive cases and 3 PR/HER2 dual positive cases in Table S2. Therefore, there are 101 TNBC patients in Table S2, which is in line with the survival curve in Fig 1 H and II (n=101). After we found this mistake, we checked throughout the data of all patients. We found the HER2 positive staining was mistakenly scored in Table S3. The pathologists use IHC staining to determine the HER2 status in our hospital. +++ is scored as positive. If it's ++ staining, additional FISH need to be done to determine the HER2 status. In original Table S3, the IHC ++ staining was all mistakenly scored as HER2 positive without referring to the FISH results. We have corrected these mistakes in the tables. We are sorry about the mistakes, and are ready to provide the original data of IHC results together with the patient information if needed.

3. Table S4 gives multivariate Cox proportional hazard analysis of 198 cases of patients with breast cancer, whilst Table S5, that of the 101 cases of TNBC patients. Line 134 – 138 however quote the data of tables S4 and S5 as suggesting that both this data are supportive that high Rac1 level is associated with chemo-resistance of breast cancers. Only the 101 TNBC patients had received neoadjuvant chemotherapy. There is no indication what sort of treatment the 198 cases had received, hence it is incorrect to use this data in support of chemo-resistance.

We agree that the 198-case cohort of all breast cancers, the 101 cases of TNBC patients and the Cox regression analysis are not supportive of chemo-resistance. They should be put together as evidence of that high Rac1 level is associated with poor outcome of breast cancer patients. Only the data of the 133 core needle biopsy patients who received neoadjuvant chemotherapy directly support the association of high Rac1 level with chemoresistance. We should be more cautious to make conclusions and have re-phased the paragraphs.

4. Based on their hypothesis, they proposed that high expression level of Rac1 contribute to resistance chemotherapy in particular DOX and DDP. It would be expected that the authors would determine the effect of Rac1 overexpression on the

sensitivity to DDP or DOX. Instead, most of the study is based on Rac1 knockdown approach, with only one experiment rather demonstrating effect of overexpression, and as supplemental data (Figure S3 H, I). The results from knockdown experiments should be supportive evidence, rather than direct evidence for the hypothesis.

Following the reviewer's suggestion, we added more experiments of overexpressing Rac1. The MDA-MB-436 cells (TNBC cancer cells) were used for these experiments because it expressed relatively low Rac1 and sensitive to DDP treatment (Figure S2A). MDA-MB-436 cells were treated with docetaxel, doxorubicin and cisplatin after Rac1 overexpression (Figure 2D). We found that Rac1 overexpression dramatically decreased the sensitivity of MDA-MB-436 cells to these chemo drugs, as shown by the increased cell proliferation (Figure 2E), colony formation (Figure 2F and S2G) and reduced apoptosis (Figure S2H, I). Consistently, the aldolase activity and the IC₅₀ of DDP increased dramatically when Rac1 was overexpressed in MDA-MB-436 cells (Figure 3G and S3E).

5. In line 143, they chose MDA-MB-231 based on the expression of Rac1 (Figure S2A). MDA-MB-231 is a triple negative breast cancer cell only. This does not mean the line is resistant to DDP. In line with the study, they should first determine the sensitivity of different breast cancer cell lines to DDP by IC₅₀ study, and show the expression of Rac1 in these cell lines. Interestingly, in Figure S2A, MCF-7 which is chemosensitive shows almost as high Rac1 expression as that of MDA-MB-231.

Once they have confirmed the Rac1 expression level and IC₅₀ to DDP, they would be able to determine the effect of Rac1 overexpression on DDP sensitivity in the low Rac1 expression cells, and conversely, determine the effect of Rac1 knockdown on DDP sensitivity in the high Rac1 expression cells.

We estimated the IC₅₀ of these cell lines (revised Figure S2A and the figure below). MDA-MB-231 cells expressed highest Rac1 among the cell lines and showed most resistance to DDP treatment. Thus, we chose MDA-MB-231 cells for the knockdown experiments. We treated MD-MBA-231 cells with the above chemotherapeutic agents after Rac1 silencing (Figure 2A). We found that siRac1 dramatically sensitized MD-MBA-231 cells to these chemo drugs, as shown by the reduced cell proliferation (Figure 2B), colony formation (Figure 2C and S2D) and increased apoptosis (Figure S2E, F). Additionally, silencing Rac1 decreased the IC₅₀ of MD-MBA-231 cells upon cisplatin treatment (Figure S3D). On the other hand, MDA-MB-436 cell expresses relatively low Rac1 and are sensitive to DDP treatment. Following the reviewer's suggestion, the phenotype experiments based on overexpression of Rac1 in MDA-MB-436 cells were performed (Figure 2D). We found that Rac1 overexpression dramatically decreased the sensitivity of MDA-MB-436 cells to these chemo drugs, as shown by the increased cell

proliferation (Figure 2E), colony formation (Figure 2F and S2G) and reduced apoptosis (Figure S2H, I). Consistently, the aldolase activity and the IC₅₀ of DDP increased dramatically when Rac1 was overexpressed in MDA-MB-436 cells (Figure 3G and S3E).

6. Line 163 states the effect of Rac1 knockdown on the IC₅₀ of DOX in MCF-7DR. However the legend of Fig S3 (A-C) states IC₅₀ of DDP on NCF-7DR cells. Is it DOX or DDP? It would seem like it should be DOX. Fig S3 (H,I) compares the IC₅₀ of DDP in MDA-MB-231 cells which overexpressed with Rac1. It would be better if they showed the effect on IC₅₀ of DDP in MDA-MB-231 for both overexpression and knockdown, since the majority of the study is based on the use of DDP. Moreover, the labeling of the subpanels of Fig S3 are largely incorrect.

We thank the review for pointing out the mistake. It is IC₅₀ of DOX, not DDP of the MCF-7DR cells in Fig S3A-C. We have corrected the mistakes in the figure legend and mislabeling in Fig S3.

Also, following the reviewer's suggestion, we have added the experiment to estimate the IC₅₀ of DDP in MDA-MB-231 cells and found that Rac1 knockdown decreased the IC₅₀ of DDP dramatically (Figure S3D).

7. In Figure 3, they determined the amount of different metabolites by mass spectrometry and identified that knockdown of Rac1 would reduce the level of various metabolites of glucose metabolism. Thus, they claimed that Rac1 could promote glycolysis and non-oxidative PPP (line 261). I agree that Rac1 can affect non-oxidative PPP thus affecting the nucleotides for DNA repair. However, their metabolites' measurement is insufficient support for the effect on glycolysis as they did not determine the rate of glucose utilization (glucose to glucose-6-phosphate, the first step in glycolysis) in their study. They merely mention their previous studies

revealed that the activation of PI3K-Rac1-cytoskeleton axis enhanced the glycolysis and non-oxidative PP through promoting aldolase A activity but do not cite any reference. Therefore, the claim is not sufficiently valid.

We are sorry to make the confusion. We have observed that PI3K-Rac1-cytoskeleton axis enhanced the glycolysis and non-oxidative PPP through promoting aldolase A activity, and published these data in *Cell* 2016 ¹ and *PNAS* 2016 ². We have cited the references in the first sentence of the paragraph instead in the middle of the paragraph.

Moreover, we added the glucose utilization experiments based on manipulating the level of Rac1 in the cell lines. The siRac1 decreased the glucose utilization in MDA-MB-231 cells (Figure 3C), while overexpression of Rac1 increased the glucose utilization in MDA-MB-436 cells (Figure 3C).

8.The effect of Rac1 in breast cancer on glycolysis and non-oxidative PPP was not clear. The mechanistic detail of Rac1 on metabolism was missing in the current study.

According to our work, Rac1 promotes glycolysis and non-oxidative PPP by regulating aldolase, as well as ERK signaling which in turn enhances the expression of enzymes of non-oxidative PPP pathway. We have explored how Rac1 regulates aldolase and promoting glycolysis and non-oxidative PPP previously ^{1, 2}. Recent studies have shown that Rac1 activation promotes cancer progression via the RAF/MEK/ERK pathway ^{3,4}. We discussed this in line 457.

Following the reviewer's suggestion, we explored how Rac1 affect ERK signaling and how ERK signaling affect the expression of enzymes of non-oxidative PPP pathway. We revealed that Rac1 mediated the activation of RAF/MEK/ERK cascade via its downstream target PAK (Figure 3J, K), which is consistent with that PAK could phosphorylate and activate RAF directly ⁵. RAF/MEK/ERK has been reported to affect glycolysis by activating MYC mediated transcription of glycolysis enzyme ^{6, 7}. Here, we focused on how the non-oxidative PPP was affected by RAF/MEK/ERK pathway. We revealed that Rac1 could up-regulated the mRNA expression of non-oxidative PPP enzymes. Therefore, we generated the luciferase vector of Rpia and Tkt and revealed that Rac1 overexpression promoted the transcriptional activity of both Rpia and Tkt promoters. The additional ERK inhibition could reverse the Rac1 mediated Rpia and Tkt transcription (Figure 3L). These data suggested that Rac1 enhanced the transcription of the non-oxidative PPP pathway enzymes via activating its downstream PAK and the subsequent RAF/MERK/ERK cascade.

9.In figures 2 and 3, they mentioned that knockdown of Rac1 would induce DNA damage by determination of γ H2AX expression. To match their hypothesis, they

should determine if overexpression of Rac1 will result in reduced DNA damage under DDP treatment.

We overexpressed Rac1 in MDA-MB-436 cells. We found that overexpression of Rac1 reduced the γ H2AX levels regardless with or without the treatment of these chemotherapeutic agents (Figure 2L) and enhanced the DNA damage repair after IR treatment (Figure S3J, L).

10. In figure 6E, the subcellular localization of Rac1 siRNA seems most unusual as majority of the siRNA appear to be localized in the cell membrane.

The *in vivo*-delivered siRNA is easy to be entrapped in the endosomes and cannot enter cytoplasm to silence target genes. Therefore, we used endosomal pH-responsive nanoparticles (NPs) to encapsulate Rac1 siRNA to enhance its endosomal escape ability and subsequent gene silencing efficacy. As shown in Figure 6E, after 4 h incubation, the loaded siRNA can escape from endosomes to the cytoplasm, as indicated by the fact that the red fluorescence (siRNA) and green fluorescence (endosomes) are not co-localized. In addition, we agree with the reviewer that it is difficult to observe the exact location of the Rac1 siRNA and some siRNAs seem to be localized on the cell membrane. In fact, similar results can be found in other published papers (ACS Nano 2011, 5, 9246-9255, Figure 3; ACS Nano 2012, 6, 4955-4965, Figure 3A; Nano Letters 2017, 17, 4427-4435; Figure 3B)^{8, 9, 10}, in which the internalized siRNA surrounds the whole cells and looks like located on the cell membrane.

We re-stained the fluorescence and took pictures by confocal as well as by bright field, which showed the endosomal escaped siRNA was in the cells but not on the cell membrane (Figure 6E).

11. In figures 7F to J, they showed bar charts but statistical analyses were missing

We added the P values accordingly.

12. The authors claim that knockdown of Rac1 could induce DNA damage (Figure 2 N-P) but these results are at most borderline. They also show the NP containing Rac1 siRNA and DDP (Figure 7B) showed no tissue specificity. Therefore, they authors should test the effect of Rac1 knockdown on cell viability of different non-cancerous cell lines to address if the treatment will have a potential to induce undesirable effect in normal human tissue. Furthermore, although they did perform mouse experiment using NP with the siRNA only, the siRNA is specific to human Rac1, therefore, the mouse model cannot determine the undesirable effect in humans.

As the reviewer mentioned, the knockdown of Rac1 induced DNA damage

(original Figure 2 N-P) were at most borderline as shown by γ H2AX levels, especially for DOX and the DTX treatment. We reformed the experiment and optimized the condition, which showed more obvious difference of γ H2AX levels (Figure 2K).

It is known that naked siRNA is rapidly cleared from blood and thus has very low tumor accumulation. In contrast, NPs have long blood circulation and preferentially accumulate at tumor sites via the enhanced permeation and retention (EPR) effect. Therefore, we employed NPs as *in vivo* Rac1 siRNA (siRac1) delivery system. We examined the biodistribution of the Cy5 labeled siRac1/DDP NPs. In comparison with naked Rac1 siRNA (siRac1), the nanoparticles containing DDP and siRac1 (siRac1/DDP NPs) show tumor specificity (Figure 7B) and the tumor accumulation of siRac1/DDP NPs was around 7-fold stronger than that of naked siRNA. On the other hand, siRac1/DDP NPs did not accumulate in muscle. The blood-rich organs, such as liver, spleen and kidney, usually showed relatively high signals of Cy5 labeled NPs. Previous reports also demonstrated the tumor specificity of NPs-based siRNA delivery systems similarly¹¹.

Following the reviewer's suggestion, we have added the cell viability of different non-cancerous cell lines upon Rac1 knockdown, including MCF-10A (normalized human mammary epithelial cell), HEK-293 (human embryonic kidney cell), MCR-5 (human lung fibroblast) cells. The Rac1 RNAi did not affect the cell viability significantly of these cells (Figure S7B).

The siRNA sequence we choose could also target the murine Rac1 (ref to the related methods). We applied the siRac1/DDP NPs to mouse breast cancer cells 4T1, and found the similar effects of suppressing Rac1 expression and inducing DNA damage (Figure 6H, I). Therefore, this siRNA was ideally to assess the therapeutic efficacy and safety of siRac1/DDP NPs in human tumor bearing mouse model. In our experiments, we have not found that siRac1/DDP NPs induced obvious systematic side effects in mice (Figure S7C-E).

13. In line 468, the authors mentioned that Rac1 is important in different human cancers. The authors should discuss whether Rac1 overexpression will be able to induce resistance to DDP in other human cancers in addition to breast cancer. If possible, they could determine the expression of Rac1 in different drug resistant cell lines in order to provide evidence to support whether Rac1 may also affect DDP sensitivity in human cancers besides breast cancer. Since Rac1 is a small GTPase, the authors should determine if any mutation will lead to constitutive activated Rac1 in human cancers.

Following this suggestion, several other cancer cells were used to test the sensitivity to platinum-based drugs after manipulating Rac1 activity. The IC₅₀ of A549CR (carboplatin resistant lung cancer cell A549) decreased upon both Rac1 siRNA or Rac1 inhibitor treatment (Figure S3F). Rac1 overexpression in SKOV-3

(ovary cancer) and ASG (gastric cancer) increased the IC50 to DDP treatment, whereas Rac1 inhibition by NSC23766 decreased the IC50 in these cells

In addition, RAC1 P29S has been reported to be a spontaneously activating cancer-associated GTPase in melanoma^{12, 13}. We cited these references in our manuscript.

Minor concern of Reviewer 3:

1. Line 31. The second word should be “Mechanistically” not mechanically.

We are sorry for the misspelling and have corrected the mistake.

2. MDA-MB-231 was written as MD-MBA-231 in few places (e.g lines 143, 144, 148..etc).

We have corrected the misspelling.

3. γ H2AX was written as γ H2ax (lines 180, 274).

We have corrected them.

4. The relation between Rac1/ERK/aldolase was not clearly described.

Our previous studies revealed that the activation of PI3K-Rac1-cytoskeleton axis enhanced the glycolysis and non-oxidative PPP through promoting aldolase A activity^{1, 2}. Aldolase has been reported as a cytoskeleton binding enzyme^{14, 15, 16}. Our studies as well as others have shown that the cytoskeleton binding of aldolase A restricts its enzymatic activity^{1, 14, 17, 18}. The cytoskeleton turnover through activation of Rac1 releases aldolase A from cytoskeleton to become the cytosolic soluble fraction^{1, 2} and thus activates aldolase A and subsequent non-oxidative PPP.

Herein, our data revealed that Rac1 activated ERK signaling by its downstream target PAK. Then, augmented ERK signaling promoted the transcription of non-oxidative PPP enzymes and thus enhanced the non-oxidative PPP activity. In Figure 3E, the ERK inhibitor decreased the level of p-ERK but not the soluble aldolase A. Therefore, the ERK signaling and aldolase are independently activated by the overexpression of Rac1. Following the reviewer’s suggestion, we have rephrased the related part of discussion to clearly describe the relation between Rac1/ERK/aldolase.

5. The authors should consider if Rac1 knockdown led to DNA damage or Rac1 knockdown compromised DNA repair. Although the outcome may be similar, but the concept is different.

We thank the reviewer for raising this question. Rac1 knockdown limited the nucleotide metabolism, which led to insufficient nucleotide supply and compromised

the DNA damage repair. We have rewritten related paragraph in the Results to clarify the concept.

References

1. Hu H, *et al.* Phosphoinositide 3-Kinase Regulates Glycolysis through Mobilization of Aldolase from the Actin Cytoskeleton. *Cell* **164**, 433-446 (2016).
2. Juvekar A, *et al.* Phosphoinositide 3-kinase inhibitors induce DNA damage through nucleoside depletion. *Proc Natl Acad Sci U S A* **113**, E4338-4347 (2016).
3. Ebi H, *et al.* PI3K regulates MEK/ERK signaling in breast cancer via the Rac-GEF, P-Rex1. *Proc Natl Acad Sci U S A* **110**, 21124-21129 (2013).
4. Kazanietz MG, Caloca MJ. The Rac GTPase in Cancer: From Old Concepts to New Paradigms. *Cancer Res* **77**, 5445-5451 (2017).
5. Baker NM, Yee Chow H, Chernoff J, Der CJ. Molecular pathways: targeting RAC-p21-activated serine-threonine kinase signaling in RAS-driven cancers. *Clin Cancer Res* **20**, 4740-4746 (2014).
6. Stefan E, Bister K. MYC and RAF: Key Effectors in Cellular Signaling and Major Drivers in Human Cancer. *Curr Top Microbiol Immunol* **407**, 117-151 (2017).
7. Stine ZE, Walton ZE, Altman BJ, Hsieh AL, Dang CV. MYC, Metabolism, and Cancer. *Cancer Discov* **5**, 1024-1039 (2015).
8. Yu H, *et al.* Overcoming endosomal barrier by amphotericin B-loaded dual pH-responsive PDMA-b-PDPA micelleplexes for siRNA delivery. *ACS Nano* **5**, 9246-9255 (2011).
9. Yang XZ, *et al.* Single-step assembly of cationic lipid-polymer hybrid nanoparticles for systemic delivery of siRNA. *ACS Nano* **6**, 4955-4965 (2012).

10. Xu X, *et al.* Tumor Microenvironment-Responsive Multistaged Nanoplatfom for Systemic RNAi and Cancer Therapy. *Nano Lett* **17**, 4427-4435 (2017).
11. Zhu X, *et al.* Long-circulating siRNA nanoparticles for validating Prohibitin1-targeted non-small cell lung cancer treatment. *Proc Natl Acad Sci U S A* **112**, 7779-7784 (2015).
12. Davis MJ, Ha BH, Holman EC, Halaban R, Schlessinger J, Boggon TJ. RAC1P29S is a spontaneously activating cancer-associated GTPase. *Proc Natl Acad Sci U S A* **110**, 912-917 (2013).
13. Olson MF. Rho GTPases, their post-translational modifications, disease-associated mutations and pharmacological inhibitors. *Small GTPases* **9**, 203-215 (2018).
14. Arnold H, Pette D. Binding of aldolase and triosephosphate dehydrogenase to F-actin and modification of catalytic properties of aldolase. *Eur J Biochem* **15**, 360-366 (1970).
15. Clarke FM, Stephan P, Huxham G, Hamilton D, Morton DJ. Metabolic dependence of glycolytic enzyme binding in rat and sheep heart. *Eur J Biochem* **138**, 643-649 (1984).
16. O'Reilly G, Clarke F. Identification of an actin binding region in aldolase. *FEBS Lett* **321**, 69-72 (1993).
17. Harris SJ, Winzor DJ. Enzyme kinetic evidence of active-site involvement in the interaction between aldolase and muscle myofibrils. *Biochim Biophys Acta* **911**, 121-126 (1987).
18. Wang J, Morris AJ, Tolan DR, Pagliaro L. The molecular nature of the F-actin binding activity of aldolase revealed with site-directed mutants. *J Biol Chem* **271**, 6861-6865 (1996).

Reviewers' comments:

Reviewer #1 (Remarks to the Author); expert in GTPases/Rac1 signalling:

The authors have addressed my concerns.

Reviewer #4 (Remarks to the Author); expert in metabolism and cancer:

The manuscript by Hu et al., described the roles of Rac1 in activating non-oxidative pentose phosphate pathway and inducing drug resistance to NAC. The authors developed an endosomal pH-responsive nanoparticle to deliver Rac1 siRNA and cisplatin to treat PDX models of breast cancer with less side effects. The authors provided significant amount of data, and made nice responses to the previous reviewers' concerns.

Reviewer #5 (Remarks to the Author); expert in Nanoparticles and cancer therapy:

The authors should consider the following points regarding the nanoparticle section of this manuscript. In addition, it is also suggested that the authors place all the original images/full scans of the WB data in the supporting information.

1. The authors indicate that their co-delivery nanoparticles (NPs) can use the characteristic of endosomal pH response to enhance the endosomal escape ability. However, the endosomal escape ability of control NPs made with non-pH-responsive polymer needs to be evaluated and compared to the current co-delivery NPs.
2. Besides co-delivery NPs, siRac1 NPs and DDP NPs are used in the in vivo experiments. However, no information about these two types of NPs can be found in the manuscript. The authors need to provide the preparation and characterizations of these NPs.
3. The results of tumor inhibition experiment show the co-delivery NPs are better than the NPs loading siRac1 or DDP. However, this better therapeutic efficacy may be due to their different pharmacokinetics (PK). Therefore, PK of the NPs loading single therapeutics should be examined and compared to the co-delivery NPs.
4. The authors announce that the co-delivery of siRac1 and DDP can synergistically inhibit tumor progression. As a synergistic treatment strategy, it is valuable to calculate the combination index. Please supplement it.
5. Scale bar is missed in Fig. 6E.
6. The authors need to double check their experimental details. For the tumor inhibition experiment, the authors indicate that three consecutive injections are administrated to the tumor-bearing mice. However, the data in Fig. 7C show only one injection.

1. The authors indicate that their co-delivery nanoparticles (NPs) can use the characteristic of endosomal pH response to enhance the endosomal escape ability. However, the endosomal escape ability of control NPs made with non-pH-responsive polymer needs to be evaluated and compared to the current co-delivery NPs.

Response: In this revised manuscript, we conducted the endosomal escape experiments using the siRNA loaded non-pH-responsive NPs, which were constructed by the commercially available polymer, Meo-PEG-*b*-PLGA. The detailed preparation has been provided in the “Methods section”.

According to the reviewer’s comment, the endosomal escape ability of the non-pH responsive NPs was evaluated. As shown in Figure 6E, after 4 h incubation, most of the siRNA loaded NPs are entrapped in the endosomes, as indicated by the co-localization of red fluorescence (siRNA) and green fluorescence (endosomes). This result strongly demonstrates that the favorable endosomal escape ability of the Meo-PEG-*b*-PDPA NPs is mainly attributed to the endosomal pH response characteristic of the Meo-PEG-*b*-PDPA polymer.

2. Besides co-delivery NPs, siRac1 NPs and DDP NPs are used in the *in vivo* experiments. However, no information about these two types of NPs can be found in the manuscript. The authors need to provide the preparation and characterizations of these NPs.

Response: In this revised manuscript, the siRac1 or DDP loaded NPs were prepared using the classic nanoprecipitation method. In brief, the Meo-PEG-*b*-PDPA polymer was dissolved in *N,N'*-dimethylformamide (DMF) to form a homogenous solution at a concentration of 10 mg/mL. For siRac1 loaded NPs, a mixture of 1 nmol siRac1 (0.1 nmol/ μ L aqueous solution) and 50 μ L of G0-C14 (5 mg/mL in DMF) were mixed with 200 μ L of Meo-PEG-*b*-PDPA solution. For DDP NPs, 10 μ L of DDP prodrug (10 mg/mL in DMF) was mixed with 200 μ L of Meo-PEG-*b*-PDPA solution. Then, under vigorously stirring (1000 rpm), the mixture was added dropwise to 5 mL of deionized water. The NP dispersion formed was transferred to an ultrafiltration device (EMD Millipore, MWCO 100 K) and centrifuged to remove the organic solvent and free compounds. After washing with PBS buffer (pH 7.4) (3×5 mL), the siRNA or DDP loaded NPs were dispersed in 1 mL of PBS buffer (pH 7.4). We have added the protocol in the related methods section (methods for preparation of NPs)

The size of the siRac1 or DDP loaded NPs was determined by dynamic light scattering (DLS) and average size of these two NPs is around 20 nm (Fig. S6). In addition, we also employed transmission electron microscope (TEM) to observe the morphology and these NPs show spherical shape (Fig. S6).

3. The results of tumor inhibition experiment show the co-delivery NPs are better than the NPs loading siRac1 or DDP. However, this better therapeutic efficacy may be due

to their different pharmacokinetics (PK). Therefore, PK of the NPs loading single therapeutics should be examined and compared to the co-delivery NPs.

Response: We have provided the PK results of siRac1 or DDP loaded NPs (Figure 7A). The half-life of these two NPs was similar to that of siRNA/DDP co-loaded NPs.

4. The authors announce that the co-delivery of siRac1 and DDP can synergistically inhibit tumor progression. As a synergistic treatment strategy, it is valuable to calculate the combination index. Please supplement it.

Response: We have calculated the combination index according to the classic method ($CI = C_{A,x}/IC_{x,A} + C_{B,x}/IC_{x,B}$), Ref. *Front. Biosci.* 2010, 2, 241-249) and the combination index of siRNA/DDP NPs was calculated 0.69 (Fig. 6H). This result indicates that the combination of RNAi therapy from siRac1 and chemotherapy from DDP shows a synergistic anti-cancer effect. We have added the calculation protocol in the related methods section (methods for combination index)

5. Scale bar is missed in Fig. 6E.

Response: We have provided the scale bar accordingly.

6. The authors need to double check their experimental details. For the tumor inhibition experiment, the authors indicate that three consecutive injections are administrated to the tumor-bearing mice. However, the data in Fig. 7C show only one injection.

Response: We have corrected the mistake and the manuscript has been also polished to make it more readable.

REVIEWERS' COMMENTS:

Reviewer #5 (Remarks to the Author):

The authors have addressed all my comments. The reviewer suggests the acceptance of the revised manuscript